

# An Aerosol Optical Depth time series 1982-2014 for atmospheric correction based on OMI and TOMS Aerosol Index

Emmihenna Jääskeläinen[1], Terhikki Manninen[1], Johanna Tamminen[1], and Marko Laine[1]

[1]Finnish Meteorological Institute, Helsinki, Finland

*Correspondence to:* Emmihenna Jääskeläinen (emmihenna.jaaskelainen@fmi.fi)

**Abstract.** The atmospheric correction of old optical satellite data is problematic, because corresponding Aerosol Optical Depth (AOD) measurements in the visible wavelength range do not exist. The construction of an AOD time series for atmospheric correction purposes to cover the period 1982-2014 is described in this paper. The AOD estimates are calculated from the Aerosol Index (AI) data from the Total Ozone Mapping Spectrometer (TOMS) and the Ozone Monitoring Instrument (OMI).

We apply this time series to the generation of the surface albedo data set CLARA-A2-SAL (the Surface ALbedo from the CM SAF cLoud, Albedo and RAdiation data set, the second version). The constructed AOD time series is temporally homogeneous, and it has sufficient quality compared to the AOD from OMI observations and from in situ measurements. The simulated atmospheric correction calculations, where the constructed AOD data are used as an aerosol input, are similar to the simulations where the aerosol information from OMI and in situ measurements is used. Also, the simulations show that the use

of the constructed AOD time series decreases the surface reflectance values (the output of the atmospheric correction) globally compared to the use of the constant AOD value 0.1.

## 1   Introduction

The surface albedo, the fraction of incoming radiation reflected hemispherically by the surface, is an essential climate variable (ECV) defined in the Implementation Plan for the Global Observing System for Climate in Support of the United Nations

Framework Convention on Climate Change (UNFCCC, http://unfccc.int/2860.php). In order to observe possible changes in the surface radiation budget, we need a data set with a long enough temporal coverage. In the Satellite Application Facility on Climate Monitoring (CM-SAF, http://www.cmsaf.eu/EN/Home/home_node.html) project, financially supported by EUMET-SAT, the 28-year long (1982-2009) broadband albedo time series CLARA-A1-SAL (the Surface ALbedo from the CM SAF cLoud, Albedo and RAdiation data set, the first version) was produced from the Advanced Very High Resolution Radiometer

(AVHRR) measurements. With this albedo data set, some major results of sea-ice zone changes in the Arctic were shown. (Schulz et al., 2009; Riihelä et al., 2013a; Karlsson et al., 2013; Riihelä et al., 2013b)

   The CLARA-A1-SAL time series use a Simplified Method for Atmospheric Correction algorithm SMAC (Rahman and Dedieu, 1994) for correcting the atmospheric effects. For the CLARA-A1-SAL data set, the aerosol optical depth (AOD), one of the main inputs of the SMAC algorithm, is fixed to have the constant value 0.1, because no global AOD time series

were available for the whole time range. Using a fixed value enables postprocessing of the CLARA-A1-SAL time series for



areas/time ranges for which accurate AOD values are available (Manninen et al., 2013). The constant value was chosen so that it is a reasonable value for most areas, but to achieve a more realistic albedo, the aerosol information must be improved (Riihelä et al., 2013a).

The objective of our study is to construct the AOD time series (1982-2014) so that we can replace the constant AOD value (0.1) with realistic AOD data for the atmospheric correction of the next version of the surface albedo time series, CLARA-A2-SAL. There are no global AOD time series that cover the whole time period of CLARA-SAL-A2, 1982-2014, but the Aerosol Index (AI) data at UV wavelength range are available from the Total Ozone Mapping Spectrometer (TOMS) and the Ozone Monitoring Instrument (OMI). Some attempts to construct the AOD information from the AI data have been made (Hsu et al., 1999; Torres et al., 2002b), but there are no daily AOD data with the global coverage needed for more accurate albedo information.

The AOD time series presented in this article is the first and robust version of it, constructed initially for the needs of the CLARA-A2-SAL data set, i.e. to remove the atmospheric contribution from the AVHHR top of atmosphere (TOA) reflectance values. This study is especially focused on the AOD values below unity, due to the restrictions of the chosen atmospheric correction algorithm of CLARA-A2-SAL. The atmospheric correction is not executed over water, because the ocean albedo retrieval of CLARA-A2-SAL does not use reflectance data at all. Hence there is no need for atmospheric correction in open water areas. Over ice and snow we assume the AOD to be small, around 0.1 (Tomasi et al., 2012), which is a commonly used assumption. So, over snow and ice the AOD information is still the constant value 0.1, but our goal for the next version of the AOD time series (version 2) is to include these areas as well. The method for calculating the AOD is based on a linear regression of AI with AOD from the OMI instrument.

Even though the AOD time series presented in this article is constructed with the CLARA-A2-SAL in mind, it can provide some usefulness to other surface quantity studies where the removal of the atmosphere contribution is necessary. For studies of the physical characteristics of the aerosols, this study might not provide any new information.

This paper is organized as follows. In Section 2 we present the input data sets used in this study, followed by Section 3 where we describe the method for constructing the AOD time series. The steps in the method are: 1) choice of regression data, 2) preprocessing of the chosen data by screening it for clouds, 3) homogenisation of the screened data and dividing it spatially using land use classification information, 4) deseasonalization of the data, 5) determining the regression function and 6) optimisation of the regression results. In Section 4 we describe the calculation of the AOD time series from TOMS-AI and OMI-AI. This includes TOMS-homogenisation and the use of spatial and temporal gap filling methods. The quality assurance of the AOD time series is carried out in Section 5. In this section we inspect the temporal homogeneity of the constructed AOD time series as well as compare the constructed data set to the OMI-AOD data of the OMAEROe product and to the in situ data. Lastly, in Section 6, we inspect the effect of the constructed AOD time series on the atmospheric correction.

The symbols and notations used in this paper are given in Table 1.



## 2  Data description

### 2.1  Data sources

TOMS instruments have been observing the aerosol index aboard the Nimbus-7, Meteor-3, Earth Probe and ADEOS satellites. Only AI retrievals from Nimbus-7 and Earth Probe measurements with TOMS are used in this study, because the AI retrievals from Meteor-3 and ADEOS TOMS measurements were not available. The Level-3 AI data from Nimbus-7 TOMS measurements provide daily, global coverage from 1978 to mid-1993. Earth Probe TOMS made measurements at the same time as ADEOS TOMS, but the Earth Probe was placed into a lower orbit than the ADEOS. Hence the Earth Probe provided higher spatial resolution at the expense of full global coverage (McPeters et al., 1998). When ADEOS TOMS closed down in December 1997, the Earth Probe satellite was boosted to a higher orbit, enabling larger daily coverage. The Level-3 AI retrievals from Earth Probe TOMS are from mid-1996 to 2005. (McPeters et al., 1996; McPeters et al., 1998)

The OMI instrument aboard the Aura satellite continues the TOMS records of aerosol index measurements. Observations are made in the afternoon, local time, and they provide nearly global daily coverage from late 2004 onwards. The OMI observations have suffered from a row anomaly problem since 2009. It varies with time and it affects the quality of the Level-1B radiance data and consequently Level-2 data products. Level-3 data used in this study are produced from filtered Level-2 data (OMI team, 2012).

The MODerate resolution Imaging Spectroradiometer (MODIS) aboard the Earth Observation System's (EOS) Terra and Aqua satellites is also used for the estimation of aerosol optical depth. Terra is on a descending orbit (southward) over the equator around 10:30 local sun time and Aqua on an ascending orbit (northward) over the equator around 13:30 local sun time. AOD estimates based on the MODIS instrument aboard the Aqua satellite are used for input data in this study because they are closer in time with the OMI instrument observations. (Hubanks et al., 2015)

The AVHRR Land Use Classification (LUC) data (Hansen et al., 1998) was generated in 1998 using AVHRR imagery acquired between 1981 and 1994. The input data spatial resolution is 1 degree. Even though the AVHRR based land use map provides accurate enough land use classification on land, it is a little bit too coarse in the coastal areas. Global Land Cover 2000 (GLC2000) provides finer spatial resolution data for the coastal areas (European Commission, Joint Research Centre, 2003).

### 2.2  Data definitions

TOMS and OMI instruments provide global daily AI data. The aerosol index is defined as

$$AI = -100 \cdot \left\{ \log_{10} \left[ \left( \frac{I_{\lambda_1}}{I_{\lambda_2}} \right)_M \right] - \log_{10} \left[ \left( \frac{I_{\lambda_1}}{I_{\lambda_2}} \right)_C \right] \right\} \tag{1}$$

where $I_{\lambda_1 M}$ and $I_{\lambda_2 M}$ are the radiances measured by instrument $M$ at wavelengths $\lambda_1$ and $\lambda_2$, and $I_{\lambda_1 C}$ and $I_{\lambda_2 C}$ are the calculated radiances, which are produced by a radiative transfer model for a pure Rayleigh atmosphere (Herman et al., 1997; Torres et al., 1998). The wavelength ranges for the AI calculation for different instruments are presented in Table 2. The TOMS data contain only positive AI values (absorbing aerosols) whereas OMI data include also negative AI values (nonabsorbing aerosols). To keep the AOD time series to be constructed homogeneous we use only positive AI values in this study. The





monthly means of the percentage of negative AI values are shown in Figure 1. From approximately 30% to 40% of the values are negative AI values and hence are discarded in the process. The geographic location of negative and positive AI values in percentages in 2008 are shown in Figure 2 (in other years, 2005-2007 and 2009-2014, the data show a similar kind behaviour). There can be seen that no areas are completely covered by negative AI values, meaning that all parts of the ground have positive

AI values for use in the process of calculating the AOD time series. The discarded aerosols are typically sea-salt particles and sulphate aerosols.

The aerosol optical depth, retrieved from OMI and MODIS measurements, is defined as

$$\tau(\lambda) = \int_0^{TOA} k_{ext}(\lambda, z)\, dz \qquad (2)$$

where $\tau(\lambda)$ is the AOD at wavelength $\lambda$, $k_{ext}$ is the wavelength depending on the aerosol extinction coefficient, which is a

measure of the attenuation of the incoming solar radiation by particle scattering $k_{sca}$ and absorption $k_{abs}$, i.e. $k_{ext} = k_{sca} + k_{abs}$ (Torres et al., 2002a). Thus, AOD is the aerosol extinction coefficient vertically integrated from the surface to the top of the atmosphere. AOD information is currently retrieved from OMI observations using two different methods: the Near-UV and the Multi-wavelength method (Torres et al., 2002a). In this study we use the OMAEROe product, retrieved using Multi-wavelength method, instead of OMAERUVd. In order to increase the AOD value reliability, in this study we use only data for which the

MODIS and OMI based AOD values at 550 nm differ from each other only by a little (subsection 3.1).

## 3 Method

The data set used for developing the method for AOD estimation using AI contains daily Level-3 OMI data of AI and AOD from 2005 to 2008. The retrievals of these years provide enough good data for method development (altogether 1454 daily maps out of 1461 possible ones). The process for finding the best fitting functions using these data sets (hereafter referred to

as the test data) can be seen in Figure 3. The preprocessing (including cloud screening, homogenisation and spatial division) is described in subsection 3.1. The prerequisites for the method are shown in subsection 3.2, and it is followed by a description of deseasonalization in subsection 3.3. In subsection 3.4 a summary of regression functions needed for the fittings is presented, and lastly, in subsection 3.5, the choice of the best fitting functions is explained.

### 3.1 Preprocessing

In the OMAEROe product OMI-AOD is reported at wavelengths 342.5 nm, 388 nm, 442 nm, 463 nm and 483.5 nm. The AOD values at 550 nm are calculated using the formula (Ångström, 1929)

$$\tau_{550} = \tau_{\lambda_{UV}} \cdot \left(\frac{550}{\lambda_{UV}}\right)^{-\alpha} \qquad (3)$$

where $\tau_{550}$ is the AOD value at 550 nm, $\tau_{\lambda_{UV}}$ is the AOD value at the wavelength $\lambda_{UV}$ and the exponent $\alpha$ is the Ångström exponent. The OMI-AOD at wavelength 550 nm (from now on $\tau_{OMI}$) is the mean value of the AOD estimates calculated using

Eq. 3 at the five wavelengths available in the OMAEROe product.





Some partly clouded pixels in the OMI data are not recognized as being cloudy. This causes an overestimation of AOD values (Torres et al., 2007), resulting in a need for removal of these pixels. This is carried out using the daily MODIS-AOD Level-3 data from which the clouded pixels are excluded. The spatial resolution difference between MODIS and OMI Level-3 data (Table 2) is resolved by manifolding the MODIS-AOD pixels to the OMI resolution. Daily $\tau_{OMI}$ and MODIS-AOD data are then compared pixelwise. If the relative difference is larger than 20%, those $\tau_{OMI}$ data pixels are masked as well as the corresponding OMI-AI pixels. We also screen AI data (and the corresponding $\tau_{OMI}$ data) simultaneously by removing the AI values that are either smaller than 0.5 (the lower limit of TOMS-AI) or larger than 4.5 (the upper limit of TOMS-AI) to homogenise the OMI-AI data levels with those of the TOMS-AI data. The resolution of OMI Level-3 data (0.25° x 0.25°) is used throughout the AOD time series calculation process (also for the resolution of final AOD time series) because it equals the CLARA-A2-SAL resolution.

To improve the data handling, the world map is divided into 65 areas on the basis of the AVHRR Land Use Classification map. This particular LUC map is chosen because the CLARA-A2-SAL time series is calculated using the AVHRR data. The AVHRR LUC contains 11 classes. Each class is manually divided into subclasses by checking how close the pixels are located to each other. An example of the division of one LUC class into subclasses is shown in Figure 4. More details are in Appendix A. In addition to facilitating the data handling, the division helps when we need to do some local inspections. Also some land cover classes and locations are related to certain AODs.

## 3.2 Prerequisites for the method

The AI is dependent on the AOD, on the height of the aerosol layer and on the absorption properties of aerosols (Torres et al., 1998). There are no global data for these quantities for the whole time period 1982-2014, only the AI data. We also want to avoid climatological assumptions as much as possible. The different correlation coefficients for each subclass for the years 2005-2008 are shown in Figure 5. The coefficients of $\tau_{OMI}$ and AI are shown in the topmost figure. There are a lot of variations, but mainly the values are around 0.3. The correlation coefficients of $\tau_{OMI}$ and the Sun Zenith Angle (SZA) are shown in the middle figure. There is not so much variation and the coefficients are mostly of the order of 0.2. The correlation coefficients of the $\tau_{OMI}$ and AI multiplied with $\cos(\theta)$, where $\theta$ is the SZA in radiances, are shown in the bottom figure. Combining the AI and SZA data improves the coefficients slightly (from around 0.3 to 0.4). The area 20 (Southern Chile, Appendix A) has small correlation coefficients in every case. It is mostly due to the fact that it has a very small amount of spatial pixels. The pixelwise correlations between AI and $\tau_{OMI}$ data from the years 2005-2008 are shown in Figure 6. Only those pixels which have more than 2 values are included. The correlation coefficients are mostly positive and mainly around 0.5. In large areas the correlation coefficient is even higher (around 0.6-0.7). These coefficients in Figures 5 and 6 show that it is possible to produce realistic AOD values to replace the constant value used thus far in the atmospheric correction of CLARA-A2-SAL, which is our main object.



### 3.3 Deseasonalization

After preprocessing, the test data of AI and AOD are deseasonalized to provide a seasonally independent relationship between the quantities. Deseasonalization is used to remove the annual variation while leaving the trend in the data. It is carried out separately for each subclass using the formula

$$z_i = y_i - (y_{mm_i} - y_{a_i}) \qquad (4)$$

where $i$ is the index of the area ($i \in [1, 65]$), $y_i$ is the original value of the pixel in the area $i$, $y_{mm_i}$ is the monthly mean (or monthly median) of the area $i$, $y_{a_i}$ is the annual mean (or annual median) of the area $i$ and $z_i$ is the deseasonalized value of the pixel in the area $i$. The coefficients $y_{mm_i}$ and $y_{a_i}$ (where $i \in [1, 65]$) are calculated as adaptive, geographically weighted averages and medians. The AI and $\tau_{OMI}$ data (in each area) are divided into bins with a length of 0.1 and the number of values in each bin is determined for both AI and $\tau_{OMI}$. For the weighted mean, the bins are first ordered by the number of values in each bin. Then the weight of the value is determined to be the number of the ordered bin (where the value belongs to) divided by the total number of bins, such that the values belonging to the bin with the largest number of counts receives the highest weight. The weighted median is calculated by constructing a vector and then adding each value to the vector as many times as the number of the values in the bin in which the value belongs to is. After this the median is calculated from the vector. Carrying out the deseasonalization either using the mean or the median values offers two slightly different data sets for choosing the best alternative for removing the annual variation.

### 3.4 Regression

The screened and deseasonalized test data are used for regression between AI and AOD after additional restrictions. The atmospheric correction algorithm SMAC is not developed to cope with AOD values larger than unity and the CLARA-A2-SAL algorithm uses only pixels for which the SZA does not exceed 70°. Thus we restrict the regression to the AI and $\tau_{OMI}$ data that satisfy AOD < 1 and SZA < 70°.

The original data are now divided into two different data sets. The first constitute the daily maps of AOD and AI for which deseasonalization is carried out using the mean values ($D_{mean}$) and the second one constitute the daily maps of AOD and AI for which deseasonalization is carried out using the median values ($D_{med}$). The relationship between the AOD and AI values is assessed pixelwise for both data sets. The actual linear regression functions have the form

$$\widetilde{\tau} = \alpha \cdot \widetilde{AI} \cdot \cos(\theta) + \beta, \qquad (5)$$

for the AOD at wavelength 550 nm, where $\widetilde{\tau}$ and $\widetilde{AI}$ denote modified AOD and AI data, respectively, and $\theta$ is the SZA. Other models were also studied but they are not shown here. This chosen model is the simplest one when taking into consideration the angle dependence in the AOD data and independence in the AI data. This study relies on statistical relationships rather than physical ones, due to the wide variety of aerosol physical characteristics and the lack of information of relative fractions of the diverse aerosol types for the time range needed.





### 3.5 Optimisation

After the determination of regression parameter values, we calculate the AOD values for years the 2005-2008 by applying the regression functions to the data sets $D_{\mathrm{med}}$ and $D_{\mathrm{mean}}$. The flow chart of this process is shown in Figure 7. These calculated AOD and the corresponding $\tau_{\mathrm{OMI}}$ values are then compared. The pixelwise correlations are examined within each subclass (65 areas). We calculate the median of the different regression coefficients $(\alpha, \beta)$ from those pixels inside the area $i$ which have a correlation $r$ higher than 0.5. Similarly, we calculate the median of the regression coefficients for pixels having $r \geq 0.6$, $r \geq 0.7$ and $r \geq 0.8$. These calculated median regression coefficients are used as fitting function coefficients. For each subclass, we have eight possible fitting functions. The forms of these functions are presented in Figure 7. For example, the coefficients $(\alpha_{0.5D_{\mathrm{med}}}, \beta_{0.5D_{\mathrm{med}}})$ from the regression of the data set $D_{\mathrm{med}}$, are calculated using the median of those pixels where $r \geq 0.5$.

For each area, one function from the eight possible ones is singled out as the final best fitting function for AI values. It is determined by calculating the AOD values for the years 2009 and 2010 which were not included in the regression parameter retrieval. To achieve data that is intercalibrated as well as possible when using both OMI-AI and TOMS-AI data, we TOMS-homogenise the OMI-AI data, i.e. we average the OMI-AI data to the TOMS-AI resolution ($1° \times 1.25°$) with restrictions $0.5 \leq$ AI $\leq 4.5$ and then manifold the data to the original OMI resolution ($0.25° \times 0.25°$). This kind of AI data from the years 2009-2010 is then deseasonalized using Eq. 4 with deseasonalization coefficients $(y_{mm_i}, y_{a_i})$, which are calculated the same way as before (subsection 3.3; adaptive, geographically weighted means and medians), but now from the TOMS-homogenised data. The AOD values are then calculated using the eight regression formulas and the resulting $\tau_{\mathrm{calc}}$ values are compared to the $\tau_{\mathrm{OMI}}$ values of the same years. We inspect the absolute differences $(\tau_{\mathrm{OMI}} - \tau_{\mathrm{calc}})$ within each subclass and determine, which regression function produces the smallest absolute differences. The chosen functions are presented in Appendix B. There $D$ stands for which deseasonalization method (mean or median) is used and $r$ stands for the correlation coefficient value that was used for determining the regression function coefficients for the area.

## 4 AOD time series construction

The best fitting functions of Appendix B are applied to the AI data from the OMI and TOMS measurements to obtain the AOD time series. The procedures of those calculations are described in the next two subsections, 4.1 and 4.2.

### 4.1 AOD time series from OMI-AI

We have OMI-AI data for the years 2005-2014. For the time series calculation these data are not screened as much as for the best fitting function calculation procedure. The reason for this is that when determining the relationship between AI and AOD we did not want outliers to dominate. On the other hand, we do allow all the natural variation to be included in the AOD data set to be generated. First the OMI-AI data are TOMS-homogenised, i.e. the data are averaged to the resolution of the original TOMS-AI data ($1° \times 1.25°$) with restrictions $0.5 \leq$ AI $\leq 4.5$, and then the new AI data are resampled to the original OMI resolution ($0.25° \times 0.25°$). The AI data are deseasonalized using Eq. 4 with the deseasonalization coefficients calculated from





the TOMS-homogenised AI data covering the years 2005-2014. The daily AOD map calculation itself is quite simple. If the daily $\overline{\text{AI}}$ map (the modified AI data map) has gaps in it, we try to fill the missing pixels using the mean from the same pixel in a temporal window of 1 day before and 1 day after the day in question. This fills the daily map sufficiently for the AOD calculation. The chosen regression functions (Appendix B) are applied to the $\overline{\text{AI}}$ and SZA data. Then the AOD values are

5 reseasonalized, meaning that the annual variation is brought back using Eq. 4 backwards. There are still gaps in the data and they are filled using the 19 x 19 weighting average matrix by location. The weight is defined to be 1/distance from the pixel to be filled. This smooths the data and fills the gaps in the daily maps. The calculation process described above is carried out for each daily OMI-$\overline{\text{AI}}$ data and it produces complete AOD day maps for the years 2005-2014. The flow chart of the process is shown in Figure 8. The monthly mean (May 2005) of the calculated AOD is shown in Figure 9, as an example.

## 4.2   AOD time series from TOMS-AI

The TOMS-AI data have almost complete global coverage during the years 1982-2004. Level-3 TOMS data have a different spatial resolution from that of the Level-3 OMI data (Table 2). The best fitting functions and area division are derived for the OMI Level-3 data, so that we need first to manifold the TOMS-AI data into the OMI resolution. After this the TOMS-AI data is screened by removing all the AI values exceeding 4.5.

There is a gap in the TOMS data in 5/7/1993 - 7/21/1996. The missing data are constructed by using the mean values of the corresponding time for the years before and after the gap for each day. We calculate the mean value pixelwise from the same date in a temporal window of 3 years before and 3 years after the gap. This is a climatological type solution for a case for which no data exist.

TOMS data have calibration problems from 2000 onwards (Kiss et al., 2007) and this affects the AI values. It is advised not

to use AI data as a proxy of aerosol-related parameters. The most obvious change is from 2002 onwards, so instead of using the TOMS-AI data from the time period 2002-2004, we treat the period as another gap of missing data. This gap is filled in the same way as the previous one.

The AI data are deseasonalized using Eq. 4 and the deseasonalization coefficients are calculated using years 1982-2004, excluding the data from the temporal gaps (mid-1993 to mid-1996, 2002-2004). The spatial gaps in the daily $\overline{\text{AI}}$ maps (the

modified AI data maps) are filled using the mean values of the daily maps in the temporal window of 1 day before and 1 day after the day in question. After the filling, we apply the chosen functions of Appendix B and then we reseasonalize the calculated AOD data using Eq. 4 backwards. The calculated AOD maps still have such large gaps in them that the weighted average matrix process cannot fill them. The solution for this is to use OMI climatology. We calculate monthly means from the calculated AOD maps based on OMI-$\overline{\text{AI}}$. The gaps in the calculated TOMS-AOD maps are filled using a spatial window of 9 x

9 pixels. Each window that has no data at all is filled with data from the calculated OMI based AOD climatology. When we do the weighted average by location (in TOMS-AOD it is 23 x 23 matrix), in the border areas where the weighting average matrix can see both climatology and calculated values, the latter ones also have a reasonable weight in the smoothing procedure, which prevents the climatology from having a too dominant effect. After the filling and smoothing, we have complete TOMS-AOD maps. This process is illustrated in Figure 10.





## 5    Quality assurance

The quality of the constructed AOD time series is inspected in several ways. In subsection 5.1 we inspect the temporal homogeneity of the time series. We also compare it to the $\tau_{OMI}$ data and to the ground-based data in subsections 5.2 and 5.3, respectively.

### 5.1    Temporal homogeneity

It is essential that the calculated AOD time series ($\hat{\tau}$ from now on) is temporally homogeneous, so that change from one satellite to another, in this case from TOMS-AI to OMI-AI, is not manifested. The monthly zonal means are shown for 33 years in Figure 11. Obviously, the annual cycle is essentially the same throughout the time series. The highest AOD values are located approximately between the Equator and the Tropic of Capricorn and they appear annually with almost the same magnitude. Similar behaviour can be seen in the other areas. The smallest AOD values appear mainly in the southern hemisphere, mainly between latitudes -50° and -60°.

The global monthly means (1982-2014), which are shown in Figure 12, confirm the temporal homogeneity. The black curves in Figure 12 are the monthly means. The orange upper and lower curves are the monthly means with added and subtracted standard deviation, respectively. The means behave similarly from year to year, as well as the standard deviation values. The cyanide curve indicates the monthly means of MODIS-AOD at wavelength 550 nm for the years 2005-2014. The upper limit of the grey area behind the curves indicates the MODIS-AOD monthly mean with added standard deviation and the lower limit the MODIS-AOD monthly mean with subtracted standard deviation. The $\hat{\tau}$ have smaller standard deviation values than the MODIS-AOD and the seasonality in the MODIS-AOD is not as distinct as it is in $\hat{\tau}$. The monthly means are mostly very similar between MODIS-AOD and $\hat{\tau}$, except in July. The higher $\hat{\tau}$ values in July are explained by the higher AOD values in Siberia. The differences in $\hat{\tau}$ and MODIS-AOD in July 2008 are shown in Figure 13 and from there can be seen the large positive difference in Siberia as well as a slightly positive difference in the north America area. The difference between $\hat{\tau}$ and MODIS-AOD in other years 2005-2007, 2009-2014 in July look similar.

### 5.2    Comparison to $\tau_{OMI}$ data

The constructed AOD time series data are compared to the $\tau_{OMI}$ data by calculating the absolute differences between the $\tau_{OMI}$ and the $\hat{\tau}$ time series data. The zonal monthly means of the absolute differences are shown in Figure 14. The negative differences show the places where the $\hat{\tau}$ have larger values than the $\tau_{OMI}$. Those differences are in the magnitude of about -0.3 at maximum. Mostly the $\hat{\tau}$ time series values are smaller than those of the $\tau_{OMI}$ (the positive values in the figure), the magnitude being mostly around 0.1-0.4.

The normalized histograms of the global seasonal AOD values from the $\tau_{OMI}$ (red) and the $\hat{\tau}$ (blue) for the years 2005-2014 are shown in Figure 15. The black vertical line indicates the constant value 0.1. Only the AOD values between 0 and unity are shown. In the seasons DJF and SON the distributions are quite similar comparing to the distributions in the seasons MAM and



JJA. In the seasons DJF and SON the peaks in the $\tau_{\text{OMI}}$ and $\hat{\tau}$ data are close to each other, whereas in the season MAM and JJA, the peaks are clearly apart. The $\hat{\tau}$ have higher AOD values globally when compared to the $\tau_{\text{OMI}}$ data in every season.

Further, we compare the $\hat{\tau}$ time series data to the $\tau_{\text{OMI}}$ data in three subclasses: subclass 1, which covers the Amazon and parts of Central America; subclass 24, which covers the areas in Mainland Southeast Asia; and subclass 39, which covers mainly the Sahara and the Middle East.

The monthly mean values of $\tau_{\text{OMI}}$ (red) and $\hat{\tau}$ (blue) for each area are shown in Figure 16. The data from the Amazon are shown in the topmost figure, the middle figure describes the data from Southeast Asia and the bottom figure describes the data from Sahara and Middle East. The $\hat{\tau}$ data from Southeast Asia and the Sahara are very similar to the $\tau_{\text{OMI}}$. In the Amazon area the peaks in the $\tau_{\text{OMI}}$ are well detected, but the $\hat{\tau}$ also produces peaks when there are none in the $\tau_{\text{OMI}}$, which is due to the nature of the AI data combined with the linear regression function. Table 3 shows the monthly mean values of the ratios between AI and $\tau_{\text{OMI}}$, the ratios between AI and $\hat{\tau}$, and the mean value of SZA, as well as the correlation coefficient between AI and $\tau_{\text{OMI}}$ from each September from the years 2005-2014. The reason for the non-existent peaks is that the regression functions were produced from the data from the years 2005-2008 when there were peaks in the $\tau_{\text{OMI}}$ data. The correlation coefficients in Table 3 are around 0.6 between $\tau_{\text{OMI}}$ and AI in the years 2005-2007 and 2010, when there are also peaks in the $\tau_{\text{OMI}}$ data. The AI data are similar year on year, but the AOD values change, which produces the peaks in the $\tau_{\text{OMI}}$ data. This can also be seen in Figure 17, in which are the means of the ratios between the AI and $\tau_{\text{OMI}}$ values (red) and between the AI and $\hat{\tau}$ values (blue) of subclass 1 in September in the years 2005-2014.

The normalized histograms of the seasonal AOD values from the $\tau_{\text{OMI}}$ (red) and the $\hat{\tau}$ (blue) data for the years 2005-2014 are shown in Figure 18. The data in the top figures are from subclass 1 (the Amazon and parts of Central America), the data in the middle ofigures are from subclass 24 (Mainland Southeast Asia) and the data in the bottom figures are from subclass 39 (the Sahara and the Middle East). The black vertical line indicates the constant value 0.1. In every area and in every season the peaks in the $\hat{\tau}$ data are higher and steeper compared to the peaks in the $\tau_{\text{OMI}}$ data. In the Amazon, the difference between the peaks in the $\tau_{\text{OMI}}$ and $\hat{\tau}$ are around 0.5 in the season JJA and over 1 in the season SON. In other seasons in the Amazon and in other areas in every season the peaks in the $\tau_{\text{OMI}}$ and $\hat{\tau}$ are close to each other, with a difference from zero (minimum) to 0.3 (maximum). In Southeast Asia the peaks in the $\tau_{\text{OMI}}$ data are close (difference 0.1 at maximum) to the constant value 0.1 in the season DJF. The same happens in the Sahara in the season DJF and SON and in the Amazon in every season. Eventhough the peaks in the $\tau_{\text{OMI}}$ data are close to the constant value 0.1, over half of the $\tau_{\text{OMI}}$ data are higher than 0.2 in almost every area and in every season.

The $\hat{\tau}$ data seem to behave similarly to the $\tau_{\text{OMI}}$ data globally and also when inspecting smaller areas. The quality of the $\hat{\tau}$ data seems to correspond well with $\tau_{\text{OMI}}$.

## 5.3 Comparison to AERONET data

AErosol RObotic NETwork (AERONET, http://aeronet.gsfc.nasa.gov/) provides globally distributed observations of spectral AOD with three data quality levels: Level 1.0 (unscreened), Level 1.5 (cloud-screened) and Level 2.0 (cloud-screened and quality-screened). In this validation we used Level 2.0 data (daily means) whenever available. If Level 2.0 was unavailable,



we used daily means of Level 1.5 data instead. The stations used in this study were chosen to be around the world and have data at least for 5 years. Instead of doing the validation by comparing directly large data pixels (0.25° x 0.25°) with in situ measurements covering tiny areas, we used a window of several pixels against several stations measurements within the window. These areas are shown in Figure 19. The chosen stations with additional information are listed in Table 4. We included

in the Figures only those months in which both data sets had at least 5 simultaneous daily means. The stations do not measure the AOD at wavelength 550 nm directly, so that value is calculated using the measured AOD values at wavelengths 440 nm and 675 nm and using Eq. 3 with the Ångström exponent calculated using AOD values at wavelengths 440 nm and 675 nm. The AOD value for each window is the mean value of these calculated AOD values from each station inside the window ($\tau_{\mathrm{AER}}$ from now on).

In Figures 20 and 21 the data window, which includes the stations GSFC, MD Science Center and Wallops, is named North America, and the data window including Alta Floresta and Rio Branco is named South America. The data window named Europe includes stations Dunkerque, Lille and Oostende, and the data window called Africa includes the stations Agoufou, IER Cinzana and Banizoumbou. The data window that includes the Beijing and XiangHe stations, is named Asia and the data window including the Tinga Tingana and Birdsville stations is named Australia.

The monthly means of $\tau_{\mathrm{AER}}$ (red) and $\widehat{\tau}$ (blue) are shown in Figure 20 for each data window. In each area, except South America, $\widehat{\tau}$ seems to be quite well in line with $\tau_{\mathrm{AER}}$. The reddish grey area in the figures indicates the maximum and the minimum of the in situ AOD at 550 nm. The $\widehat{\tau}$ stay almost all the time and in every inspection window between those limits.

    The normalized histograms of the seasonal AOD values from $\tau_{\mathrm{AER}}$ (red) and $\widehat{\tau}$ (blue) for the different data windows are shown in Figure 21. The black vertical line indicates the constant value 0.1. The AOD data axis is limited to the range $[0, 1]$ to

20 better show the features in that range. Generally, the distributions are either quite well in line with each other in different areas and seasons or there are slightly shifted (around 0.1) in $\widehat{\tau}$ values towards to the higher AOD values. So $\widehat{\tau}$ overestimates AOD values. Also, the peaks in the $\widehat{\tau}$ are steeper.

## 6   The effect of $\widehat{\tau}$ on the atmospheric correction

We performed simulations with the atmospheric correction algorithm SMAC. The inputs needed by SMAC are shown in Table

5, along with the constant values assumptions. The output of SMAC is a surface reflectance. The pixelwise median values of the relative differences (magnitude values) of surface reflectances for the year 2010 are shown in Figure 22. The relative difference between surface reflectance calculated using $\tau_{\mathrm{OMI}}$ as AOD information and surface reflectance calculated using $\widehat{\tau}$ as input AOD values are shown in the top figure, and the differences between surface reflectance calculated using $\tau_{\mathrm{OMI}}$ and surface reflectance calculated using the constant AOD value 0.1 are shown in the bottom figure. The grey colour indicates the

areas without data. In these simulations the top of atmosphere reflectance is assumed to be 0.1 and the satellite zenith angle 40°. The differences are smaller when using $\widehat{\tau}$ instead of the constant value 0.1, especially in Amazon, west Africa, India and southeast Asia where the relative difference between SMAC[$\tau_{\mathrm{OMI}}$] and SMAC[$\widehat{\tau}$] is from 20% to the 70% compared to the





difference between SMAC[$\tau_{OMI}$] and SMAC[0.1] (from 60% to 100% and over). The mean magnitude relative difference is 31.0% between SMAC[$\tau_{OMI}$] and SMAC[$\hat{\tau}$] and 44.2% between SMAC[$\tau_{OMI}$] and SMAC[0.1].

The same simulations but without magnitude values are shown in Figure 23. The relative differences between SMAC[$\tau_{OMI}$] and SMAC[0.1] are negative almost globally, from -20% to -80%. The relative differences between SMAC[$\tau_{OMI}$] and SMAC[$\hat{\tau}$]

vary more between negative and positive values, from around -30% to 20%.

Figure 24 shows the surface reflectance values simulated by SMAC for subclasses 1, 24 and 39 (the Amazon and parts of Central America, Mainland Southeast Asia and the Sahara and the Middle East, respectively) for each month in the year 2006. The assumptions used for constant values for the satellite zenith angle are 0° (black colour) and 40° (magenta) and for ToA reflectance 0.05 (circle), 0.1 (asterisk) and 0.15 (triangle). The Solar zenith angles, Solar azimuth angles and AOD at 550 nm for

each month are chosen to be the most probable value from each area of the year 2006. The comparisons between SMAC[$\tau_{OMI}$] and SMAC[$\hat{\tau}$] are shown in the top figures, and the comparisons between SMAC[$\tau_{OMI}$] and SMAC[0.1] are shown in the bottom figures. In Southeast Asia and the Sahara the surface reflectance values in the top figures are well in line compared to the surface reflectances calculated by using SMAC[$\tau_{OMI}$] and SMAC[0.1]. The surface reflectance values calculated by using SMAC[0.1] tend to be higher than the SMAC[$\tau_{OMI}$] values. In the Amazon, both inspections look similar.

The surface reflectances are also simulated by SMAC for three AERONET data windows: South America, Africa and Asia for each month, and are shown in Figure 25. The needed SMAC inputs, which are not set as constant, are calculated by choosing the most probable value in each month from the whole time period for which there are data from AERONET (Table 4). The used constant assumptions are the same as in the inspections with $\tau_{OMI}$. The comparisons between SMAC[$\tau_{AER}$] and SMAC[$\hat{\tau}$] are shown in the top figures, and the comparisons between SMAC[$\tau_{AER}$] and SMAC[0.1] are shown in the bottom

figures. There can be seen the same kind of behaviour as in Figure 24. In Africa and Asia the surface reflectances calculated by SMAC[$\hat{\tau}$] are more in line with SMAC[$\tau_{AER}$] than the surface reflectance values calculated by SMAC[0.1], which tend to be higher than the SMAC[$\tau_{AER}$] values. In South America the surface reflectances from SMAC[$\tau_{AER}$] and SMAC[0.1] are better in line than the surface reflectances from SMAC[$\tau_{AER}$] and SMAC[$\hat{\tau}$]. Finally, when comparing the surface reflectance values from SMAC[$\hat{\tau}$] and from SMAC[0.1] with SMAC[$\tau_{OMI}$] and SMAC[$\tau_{AER}$] values, it can be seen that using the $\hat{\tau}$ as an AOD

input in the SMAC algorithm will mostly decrease the surface reflectance values.

## 7   Conclusions

The main object was to produce an AOD time series that would be realistic from the point of view of atmospheric correction. The AOD time series from 1982 to 2014 at wavelength 550 nm was constructed from the aerosol index data at UV wavelength range retrieved by TOMS and OMI instruments, together with SZA information. The resulting AOD time series is temporally

homogeneous throughout the years 1982-2014 and the comparison with the OMI-AOD at 550 nm and the in situ AOD at 550 nm from AERONET measurements shows that the quality of the constructed AOD time series is sufficient for its intended use. The atmospheric correction simulations (calculated using SMAC) show that the use of the constructed AOD time series produces similar surface reflectance values as the use of the OMI-AOD or the in situ AOD values as the AOD information.





# Appendices

## A The land use classification information

Descriptions and locations of land use subclasses are presented in Tables A1, A2 and A3.

## B The information of best fitting functions

5  The chosen coefficients $\alpha, \beta$ for each area are presented in Table B1.

*Acknowledgements.* This work was funded by the EUMETSAT Satellite Application Facility on Climate Monitoring (CM SAF). Work of Johanna Tamminen and Marko Laine was funded by Finnish Academy project INQUIRE. The authors would like to thank the NASA data centers for providing AOD and AI data from the OMI and TOMS instruments, as well as providing AOD data from the MODIS instrument. The authors would like to thank Global Land Cover Facility for providing AVHRR Land Use Classification data and the European

10  Commission Joint Research Centre for providing the GLC2000 Land Use Classification data. The authors are thankful to AERONET for the quality-controlled in situ AOD data used in this study. The authors want to thank Piet Stammes (KNMI) for his constructive comments. The authors would also like to thank Kaj Andersson (VTT), Aku Riihelä and Niilo Kalakoski (Finnish Meteorological Institute) for helpful discussions.



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



**Table 1.** The symbols and notations which are used in this study.

| Symbol/notation | Description | Sections |
|---|---|---|
| OMI-AOD | AOD retrieved from OMI observations | 3.1 |
| MODIS-AOD | AOD retrieved from MODIS observations | 3.1; 5.1 |
| $\tau_{OMI}$ | the mean value of the AOD estimates at wavelength 550nm | 3; 5.2; 6 |
| $\tau_{AER}$ | the mean value of calculated AOD values (from AERONET data) at 550 nm | 5.3; 6 |
| $\tau_{calc}$ | calculated AOD data for testing | 3.5 |
| $\hat{\tau}$ | constructed AOD time series 1982-2014 | 5; 6 |
| $\tilde{\tau}$ | modified (preprocessed, deseasonalized) AOD data | 3.4 |
| OMI-AI | AI retrieved from OMI observations | 3.1; 3.5; 4.1; 5.1 |
| TOMS-AI | AI retrieved from TOMS observations | 3.1; 3.5; 4.1; 4.2; 5.1 |
| OMI-$\overline{AI}$ | modified (screened, TOMS-homogenised, deseasonalized) AI data | 4 |
| TOMS-$\overline{AI}$ | modified (screened, manifolded, deseasonalized) AI data | 4.2 |
| $\widetilde{AI}$ | modified (preprocessed, deseasonalized) AI data | 3.4 |
| $D_{mean}$ | the data set of AOD and AI daily maps, the deseasonalization is carried out using mean values | 3.4; 3.5 |
| $D_{med}$ | the data set of AOD and AI daily maps, the deseasonalization is carried out using median values | 3.4; 3.5 |
| SMAC[X] | a surface reflectance values calculated by SMAC, X corresponds to the used AOD information | 6 |

**Table 2.** Properties of the instruments, whose data we used in this study.

| | Satellite | Product | Version | Period | AI | AOD | AI calculation | L3 resolution |
|---|---|---|---|---|---|---|---|---|
| TOMS | Nimbus-7 | TOMSN7L3 | v8 | 11/1978 - 05/1993 | x | - | 340.0 nm - 380.0 nm | 1.00° x 1.25° |
| | Earth Probe | TOMSEPL3 | v8 | 07/1996 - 12/2005 | x | - | 331.0 nm - 360.0 nm | 1.00° x 1.25° |
| OMI | Aura | OMAEROe | v003 | 08/2004 - | x | x | 342.5 nm - 388.0 nm | 0.25° x 0.25° |
| MODIS | Aqua | MYD08 | 005 (2005-2008) 051 (2009-2013) 006 (2014) | 12/1999 - | - | x | - | 1.00° x 1.00° |



**Table 3.** The monthly mean values of the ratios between the AI and $\tau_{OMI}$ values (m($\tau_{OMI}$/AI)), the ratios between the AI and $\hat{\tau}$ (m($\hat{\tau}$/AI)), and the mean value of SZA as well as the correlation coefficient between the AI and $\tau_{OMI}$ values (r) from each September from the years 2005-2014 of subclass 1 (Amazon and parts of Central America).

| year | m($\tau_{OMI}$/AI) | m($\hat{\tau}$/AI) | m(SZA) | r |
|------|------|------|------|------|
| 2005 | 0.78 | 1.13 | 30.5° | 0.63 |
| 2006 | 0.78 | 1.04 | 31.2° | 0.60 |
| 2007 | 0.80 | 1.24 | 31.0° | 0.65 |
| 2008 | 1.21 | 0.68 | 31.0° | 0.47 |
| 2009 | 5.48 | 0.78 | 29.4° | 0.26 |
| 2010 | 0.98 | 1.11 | 30.7° | 0.55 |
| 2011 | 1.55 | 0.80 | 30.7° | 0.41 |
| 2012 | 1.61 | 0.88 | 31.0° | 0.42 |
| 2013 | 2.42 | 0.67 | 30.9° | 0.12 |
| 2014 | 3.59 | 0.81 | 29.8° | -0.04 |



**Table 4.** Description of the chosen AERONET stations. The period for which the data are available and the information of which years of the data is Level 1.5 instead of Level 2.0 are shown.

| Name | (lat, lon) | LUC | Period | Level 1.5 data |
|------|-----------|-----|--------|----------------|
| GSFC (North America) | (38.99, -76.83) | Wooded Grassland | 1999-2014 | - |
| MD Science Center (North America) | (37.94, -75.48) | Mixed Coniferous Forest and Woodland | 1999-2014 | - |
| Wallops (North America) | (39.28, -76.62) | Wooded Grassland | 1999-2014 | - |
| Alta Floresta (South America) | (-9.87, -56.10) | Broadleaf Evergreen Forest | 2000-2013 | - |
| Rio Branco (South America) | (-9.96, -67.87) | Broadleaf Evergreen Forest | 2000-2013 | - |
| Dunkerque (Europa) | (51.04, 2.37) | Cultivated Crops | 2003-2014 | - |
| Lille (Europa) | (50.61, 3.14) | Wooded Grassland | 2003-2014 | - |
| Oostende (Europa) | (51.23, 2.93) | Cultivated Crops | 2003-2014 | - |
| Agoufou (Africa) | (15.35, -1.48) | Grassland | 2004-2011 | 2010-2011 |
| Banizoumbou (Africa) | (13.54, 2.66) | Shrubs and Bare Ground | 2004-2011 | - |
| IER Cinzana (Africa) | (13.28, -5.93) | Grassland | 2004-2011 | - |
| Beijing (Asia) | (39.98, 116.38) | Broadleaf Decidious Forest and Woodland | 2001,2004-2014 | - |
| XiangHe (Asia) | (39.75, 116.96) | Broadleaf Decidious Forest and Woodland | 2001,2004-2014 | - |
| Birdsville (Australia) | (-25.90, 139.34) | Shrubs and Bare Ground | 2005-2012 | - |
| Tinga Tingana (Australia) | (-28.98, 139.99) | Shrubs and Bare Ground | 2005-2012 | - |


**Table 5.** The inputs needed by the atmospheric correction algorithm SMAC.

| Input | value if constant |
|---|---|
| Solar zenith angle (degrees) | - |
| Solar azimuth angle (degrees) | - |
| Satellite zenith angle (degrees) | 0°, 40° |
| Satellite azimuth angle (degrees) | 260° |
| Water vapor content (g/cm$^2$) | 2.5 |
| Integrated ozone (atm/cm$^2$) | 0.35 |
| Aerosol optical depth at 550 nm | - |
| Pressure at surface level (hPa) | 1013 |
| TOA reflectance ( 0 .. 1 ) | 0.05, 0.1, 0.15 |





**Table A1.** The land use classification information.

| Land Use Classification | | Subclass | Location or description |
|---|---|---|---|
| 1 | Broadleaf Evergreen Forest | 1 | Amazon and parts of Central America |
| | | 2 | Congo River Basin and Madagascar Rainforests |
| | | 3 | Rainforests of Southeast Asia |
| 2 | Coniferous Evergreen Forest and Woodland | 4 | North America Taiga |
| | | 5 | Eurasia Taiga |
| 3 | High Latitude Deciduous Forest and Woodland | 6 | Transitional zone of North America Taiga and Tundra |
| | | 7 | Transitional zone of Eurasia Taiga and Tundra |
| 4 | Tundra | 8 | North America Tundra |
| | | 9 | Eurasia Tundra |
| 5 | Mixed Coniferous Forest and Woodland | 10 | On the east coast of North America |
| | | 11 | Central Europe |
| | | 12 | Japan and small areas on the east coast of Asia |
| | | 13 | Small areas in the middle of Chile and in the Northern Argentina |
| | | 14 | Small areas in the southeast coast of Australia |
| | | 15 | Small areas in the southern Africa |
| 6 | Wooded Grassland | 16 | Parts of west Coast of North America |
| | | 17 | Areas in the southeast of USA and parts of Central America |
| | | 18 | Areas on the North and northwest coast of South America |
| | | 19 | Savanna area in the South America |
| | | 20 | Southern Chile |
| | | 21 | Parts of west Europe |
| | | 22 | Savanna area below the Sahel in Africa |
| | | 23 | Savanna area in the southern Africa |
| | | 24 | Areas in the Mainland Southeast Asia |
| | | 25 | Areas in the Maritime Southeast Asia |
| | | 26 | southeast coast of Australia |



**Table A2.** The land use classification information.

| Land Use Classification | | Subclass | Location or description |
|---|---|---|---|
| 7 | Grassland | 27 | Southern coast of Alaska |
| | | 28 | South America prairies |
| | | 29 | Small areas on the northern coast of South America |
| | | 30 | On the southern part of the west coast of South America |
| | | 31 | Areas in the northern and southern Argentina |
| | | 32 | Southern Iceland and small areas in the northern Europe |
| | | 33 | Steppes of central Asia and western coast of India |
| | | 34 | Sahel |
| | | 35 | Large areas in the southern Africa (Steppe area of southern Africa) |
| | | 36 | Mostly the north coast of Australia, small areas in the southern Australia |
| 8 | Bare Ground | 37 | Small areas in the western USA |
| | | 38 | Southern coast of Peru and northern Chile |
| | | 39 | Sahara and Middle East |
| | | 40 | Gobi desert |
| | | 41 | Namib desert |
| | | 42 | Small area in the middle of Australia |
| 9 | Shrubs and Bare Ground | 43 | Areas in the western USA |
| | | 44 | Most of the southern Argentina, small parts of Chile and Peru |
| | | 45 | Areas around the Sahara and parts in the middle Asia |
| | | 46 | Kalahari desert |
| | | 47 | Desert area in the Australia |





**Table A3.** The land use classification information.

| Land Use Classification | | Subclass | Location or description |
|---|---|---|---|
| 10 | Cultivated Crops | 48 | Large area in the middle of the North America |
| | | 49 | Small areas in the Mexico and in the southern part of Central America |
| | | 50 | Small areas along the east coastal line of South America |
| | | 51 | Most of the Central Europa |
| | | 52 | Small areas around Sahel and small areas in the southern Africa |
| | | 53 | Eastern China |
| | | 54 | Most of the India |
| | | 55 | Small areas in the Mainland and Maritime Southeast Asia |
| | | 56 | Areas around the desert area in the Australia |
| 11 | Broadleaf Deciduous Forest and Woodland | 57 | Small areas in the western North America |
| | | 58 | Small areas in the eastern North America |
| | | 59 | Small areas in the middle of the South America |
| | | 60 | Small areas in the Central Europe |
| | | 61 | Small areas in the southern Africa |
| | | 62 | Small areas on the eastern coast of Russia |
| | | 63 | Small areas in the Mainland Southeast Asia |
| | | 64 | Small areas in the Maritime Southeast Asia and in the Australia |
| - | - | 65 | New Zealand |



**Table B1.** The information of the best fitting functions chosen for each subclass.

| Area | $\alpha$ | $\beta$ | D | r | Area | $\alpha$ | $\beta$ | D | r |
|------|--------|--------|--------|-----|------|--------|--------|--------|-----|
| 1 | 0.325 | 0.339 | median | 0.5 | 34 | 0.166 | 0.190 | mean | 0.6 |
| 2 | 0.209 | 0.264 | median | 0.5 | 35 | 0.261 | 0.096 | mean | 0.5 |
| 3 | 0.307 | 0.217 | median | 0.5 | 36 | 0.095 | 0.052 | median | 0.5 |
| 4 | 0.367 | 0.088 | median | 0.5 | 37 | 0.135 | 0.197 | mean | 0.5 |
| 5 | 0.313 | 0.110 | median | 0.6 | 38 | 0.044 | 0.210 | median | 0.5 |
| 6 | 0.102 | 0.515 | mean | 0.7 | 39 | 0.189 | 0.119 | mean | 0.5 |
| 7 | 0.578 | 0.109 | median | 0.5 | 40 | 0.249 | 0.156 | median | 0.5 |
| 8 | 0.112 | 0.125 | median | 0.5 | 41 | 0.127 | 0.119 | mean | 0.6 |
| 9 | 0.208 | 0.095 | median | 0.5 | 42 | 0.060 | 0.088 | mean | 0.7 |
| 10 | 0.047 | 0.133 | mean | 0.5 | 43 | 0.144 | 0.126 | mean | 0.8 |
| 11 | 0.081 | 0.160 | mean | 0.5 | 44 | 0.048 | 0.145 | median | 0.5 |
| 12 | 0.164 | 0.328 | mean | 0.6 | 45 | 0.183 | 0.176 | mean | 0.6 |
| 13 | 0.126 | 0.214 | mean | 0.5 | 46 | 0.157 | 0.098 | median | 0.5 |
| 14 | 0.144 | 0.084 | mean | 0.5 | 47 | 0.038 | 0.079 | mean | 0.8 |
| 15 | 0.317 | 0.089 | mean | 0.8 | 48 | 0.130 | 0.138 | mean | 0.5 |
| 16 | 0.186 | 0.113 | mean | 0.5 | 49 | 0.099 | 0.141 | mean | 0.8 |
| 17 | 0.034 | 0.134 | mean | 0.7 | 50 | 0.105 | 0.110 | mean | 0.5 |
| 18 | 0.168 | 0.254 | mean | 0.5 | 51 | 0.151 | 0.163 | mean | 0.5 |
| 19 | 0.212 | 0.140 | median | 0.5 | 52 | 0.202 | 0.198 | mean | 0.5 |
| 20 | -0.015 | 0.068 | median | 0.8 | 53 | 0.458 | 0.297 | median | 0.5 |
| 21 | 0.161 | 0.141 | mean | 0.5 | 54 | 0.173 | 0.298 | mean | 0.7 |
| 22 | 0.247 | 0.131 | median | 0.5 | 55 | 0.066 | 0.313 | median | 0.7 |
| 23 | 0.356 | 0.113 | mean | 0.8 | 56 | 0.085 | 0.081 | mean | 0.5 |
| 24 | 0.287 | 0.215 | mean | 0.8 | 57 | 0.049 | 0.145 | mean | 0.8 |
| 25 | 0.092 | 0.138 | median | 0.5 | 58 | 0.092 | 0.139 | mean | 0.6 |
| 26 | 0.106 | 0.080 | mean | 0.5 | 59 | 0.182 | 0.222 | median | 0.5 |
| 27 | -0.187 | 0.199 | median | 0.5 | 60 | 0.197 | 0.146 | mean | 0.7 |
| 28 | 0.110 | 0.136 | mean | 0.5 | 61 | 0.210 | 0.179 | mean | 0.6 |
| 29 | 0.118 | 0.268 | mean | 0.5 | 62 | 0.112 | 0.319 | mean | 0.8 |
| 30 | 0.061 | 0.445 | median | 0.7 | 63 | 0.170 | 0.317 | mean | 0.5 |
| 31 | 0.063 | 0.113 | median | 0.7 | 64 | 0.087 | 0.101 | mean | 0.5 |
| 32 | 0.089 | 0.111 | mean | 0.5 | 65 | 0.021 | 0.052 | median | 0.5 |
| 33 | 0.205 | 0.160 | mean | 0.8 | - | - | - | - | - |




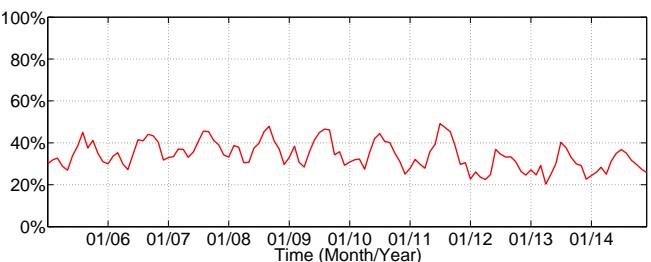

**Figure 1.** The percentage of negative AI values globally in each month in the years 2005-2014.

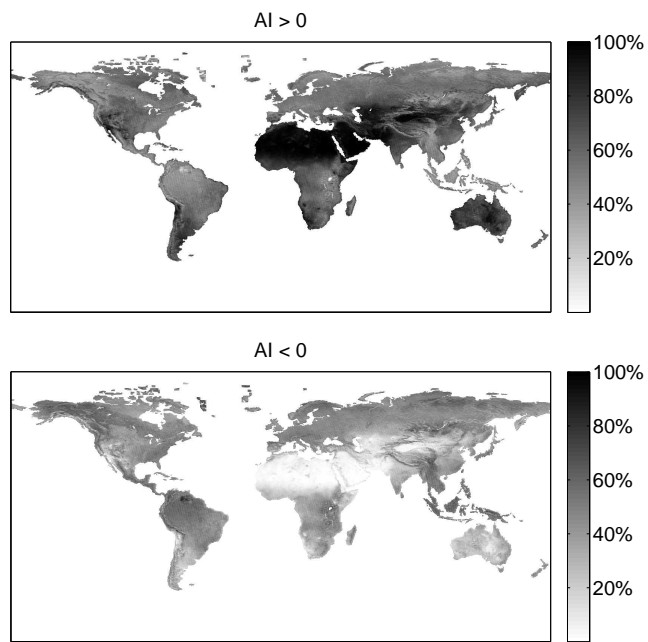

**Figure 2.** The percentage of positive (above) and negative (below) AI values globally in the year 2008.





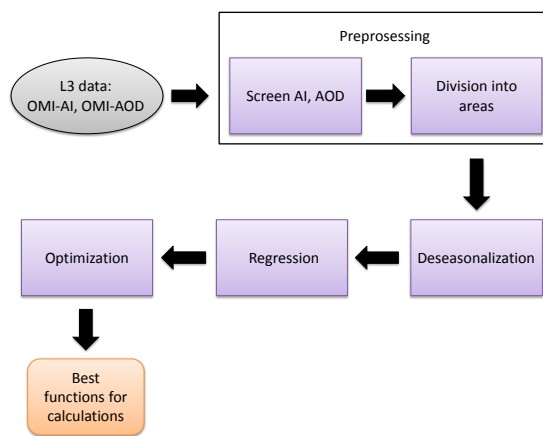

**Figure 3.** Flow chart of the procedure of finding the best functions for AOD time series.

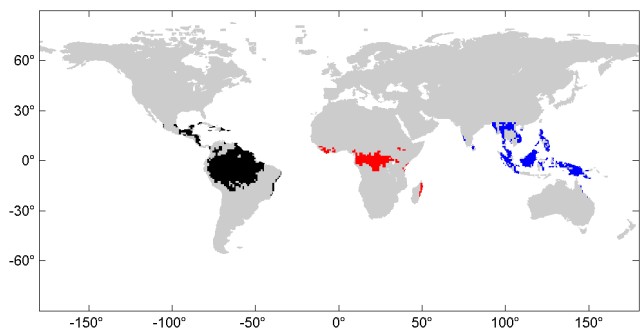

**Figure 4.** An example of data division using the AVHRR LUC data. Coloured areas (black, red and blue) all have the land use classification value 1 (Broadleaf Evergreen Forest). Based on the location, the class 1 is divided into 3 areas (marked by the different colours), which are treated separately in processing the AOD time series (Appendix A).





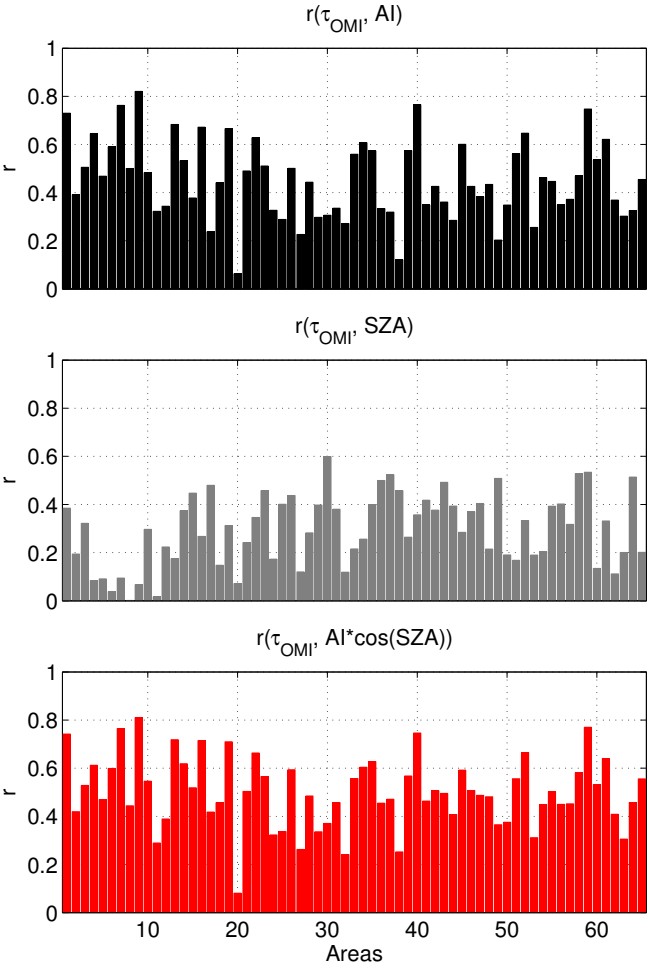

**Figure 5.** The correlation coefficients of AOD and AI (topmost), of AOD and SZA (middle) and of AOD and AI data multiplied with cos(SZA) (bottom) of different areas for the years 2005-2008.

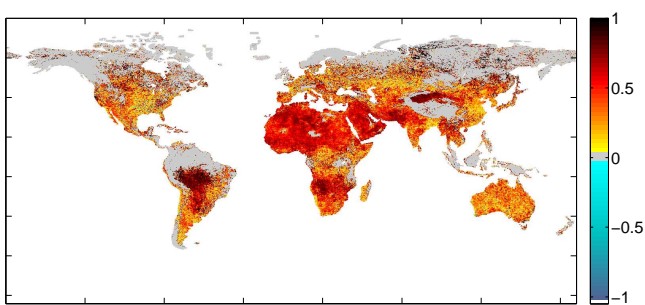

**Figure 6.** The correlation coefficients of AOD and AI pixelwise from the years 2005-2008. The grey colour indicates the area without data.





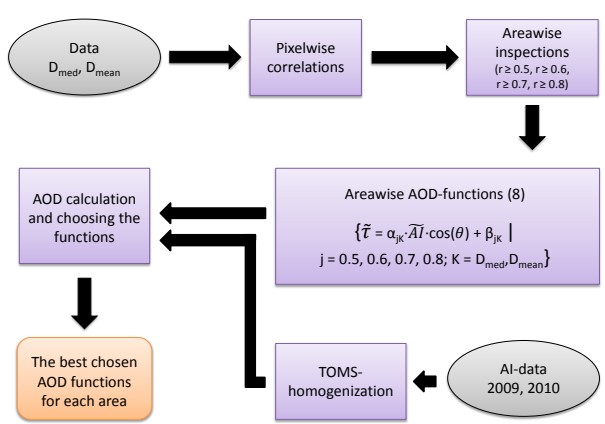

**Figure 7.** Flow chart of the process of choosing the best functions.

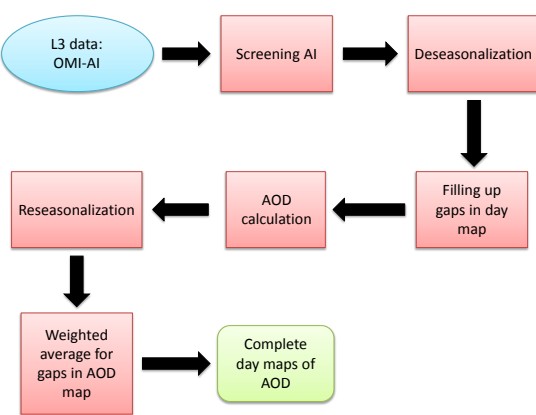

**Figure 8.** Flow chart of the process of calculating the AOD maps using OMI-AI and SZA information.



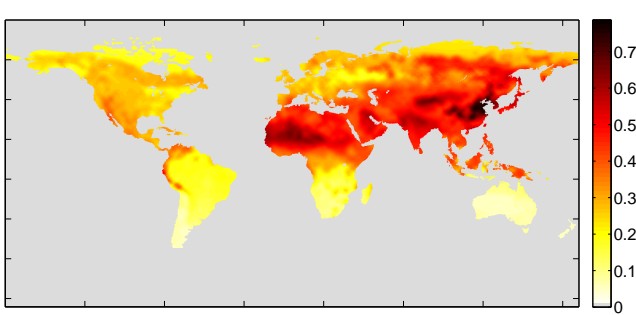

**Figure 9.** The monthly mean AOD map of May 2005, calculated from the OMI-AI data.

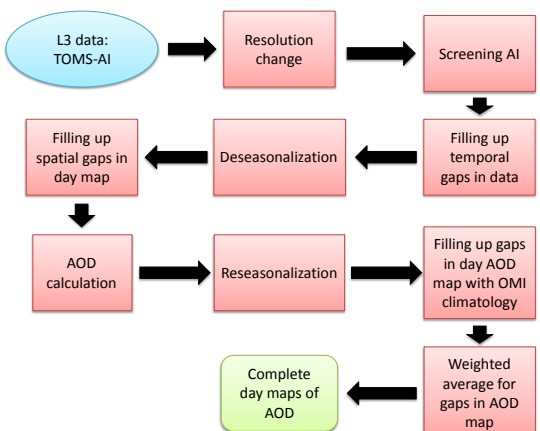

**Figure 10.** Flow chart describing the calculation process of the AOD maps using TOMS-AI and SZA information.



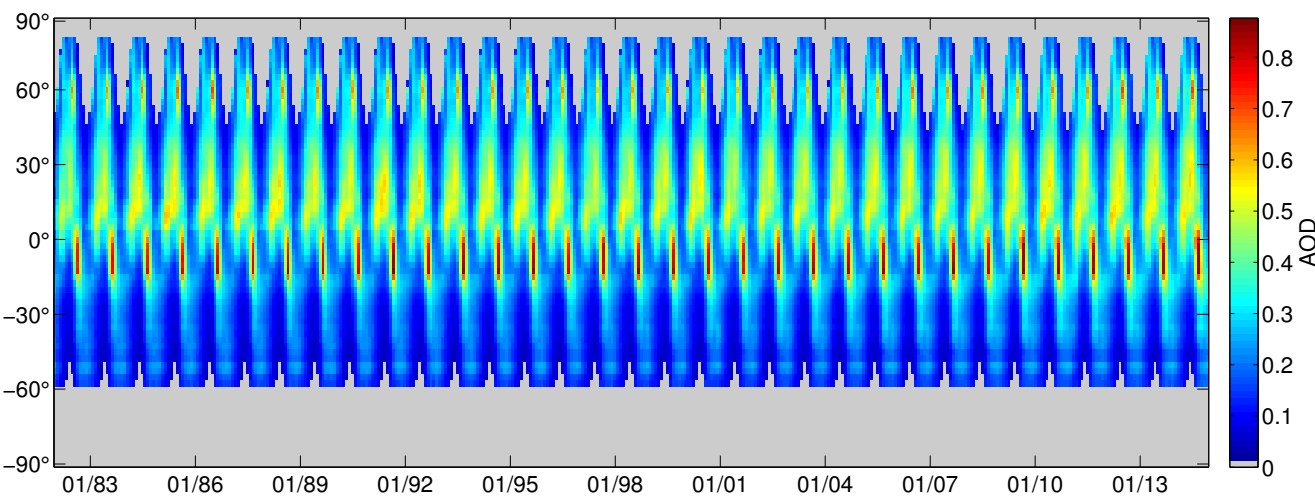

**Figure 11.** The monthly zonal means of the $\hat{\tau}$ time series, covering the years 1982-2014. The grey colour indicates the months without data.

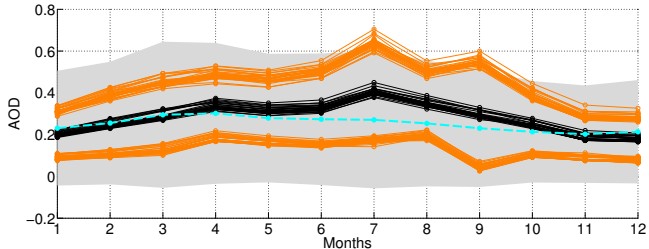

**Figure 12.** The global monthly means of the whole $\hat{\tau}$ time series. The black curves correspond to the monthly means. The upper and lower orange curves indicate the corresponding monthly means with standard deviations added and subtracted. The cyanide curve corresponds to the monthly mean of MODIS-AOD at wavelength 550 nm for the years 2005-2014 and the upper and lower limit of the grey area indicate the MODIS-AOD monthly mean with standard deviations added and subtracted.

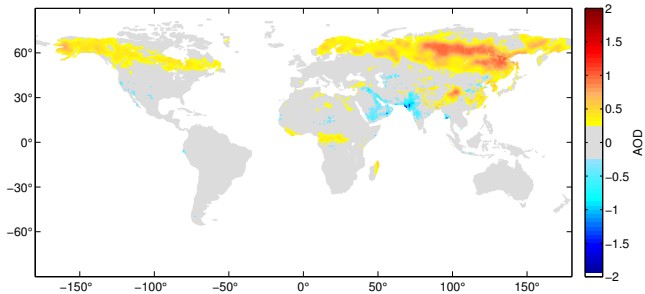

**Figure 13.** The difference between $\hat{\tau}$ and MODIS-AOD in July 2008. The positive difference indicates higher $\hat{\tau}$ values, the negative difference the opposite. Small differences in a range [-0.25, 0.25] are omitted (grey).




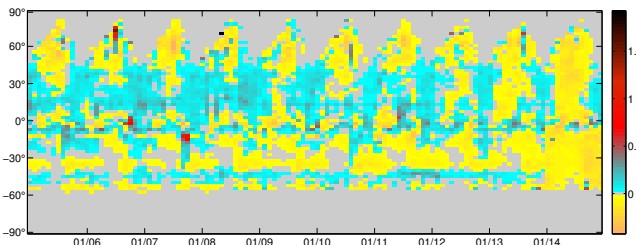

**Figure 14.** The zonal monthly means of absolute differences between the $\tau_{OMI}$ and the $\hat{\tau}$ time series data. The negative values show the areas where the $\hat{\tau}$ values are larger than the $\tau_{OMI}$ whereas the positive values show the areas where the opposite takes place. The grey colour indicates the months without data.

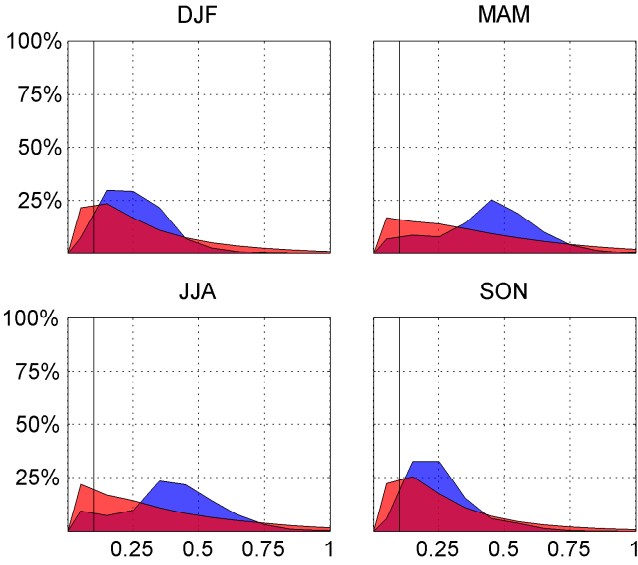

**Figure 15.** The normalized histograms of the seasonal AOD values (global) from the $\tau_{OMI}$ (red) and the $\hat{\tau}$ (blue) for the years 2005-2014. The black vertical line indicates the constant value 0.1.




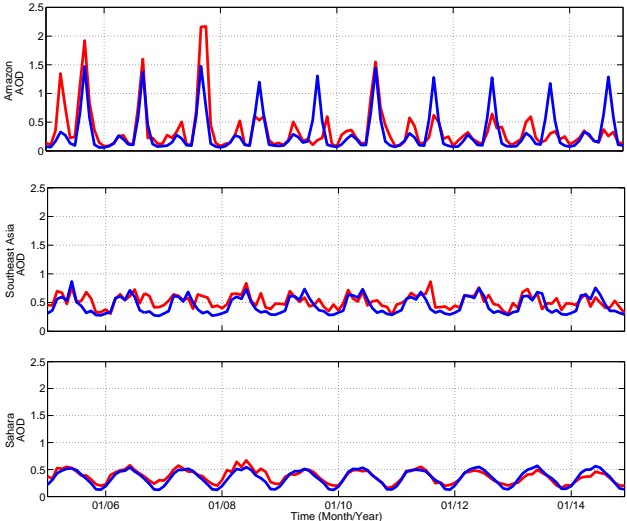

**Figure 16.** The monthly mean values of the $\tau_{OMI}$ (red) and the $\hat{\tau}$ (blue) data. The data in top figure are from subclass 1 (the Amazon and parts of Central America), the data in the middle one from subclass 24 (Mainland Southeast Asia) and the data in the bottom figure are from subclass 39 (the Sahara and the Middle East).

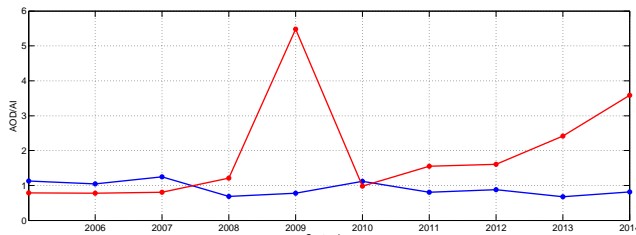

**Figure 17.** The mean values of the ratios between AI and $\tau_{OMI}$ values (red) and between AI and $\hat{\tau}$ values (blue) of subclass 1 (Amazon and parts of Central America) in Septembers for the years 2005-2014.




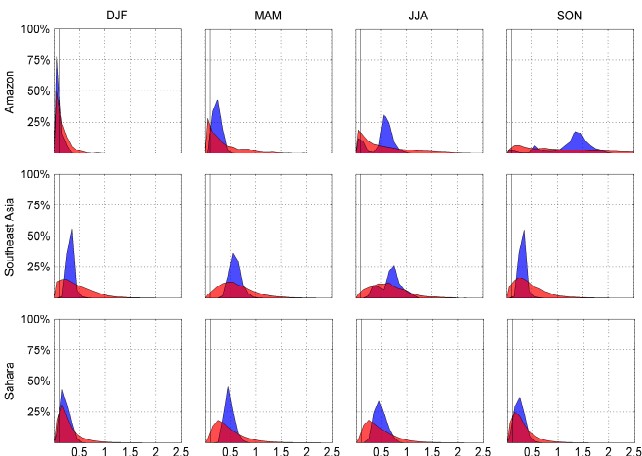

**Figure 18.** The normalized histograms of the seasonal AOD values from the $\tau_{\mathrm{OMI}}$ (red) and the $\hat{\tau}$ (blue) data for the years 2005-2014. The data in top figures are from subclass 1 (the Amazon and parts of Central America), the data in the middle ones from subclass 24 (Mainland Southeast Asia) and the data in the bottom figures are from subclass 39 (the Sahara and the Middle East). The black vertical line indicates the constant value 0.1.

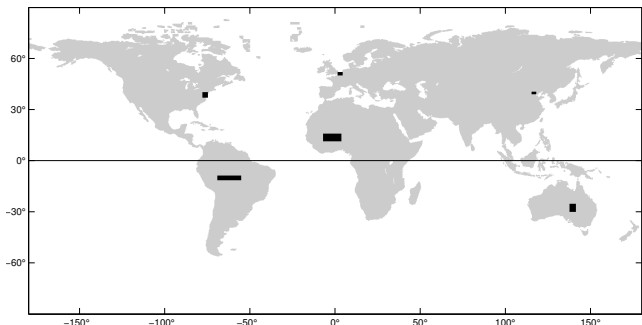

**Figure 19.** The chosen six inspection windows for quality assurance (black boxes) which contain at least 2 AERONET stations each.





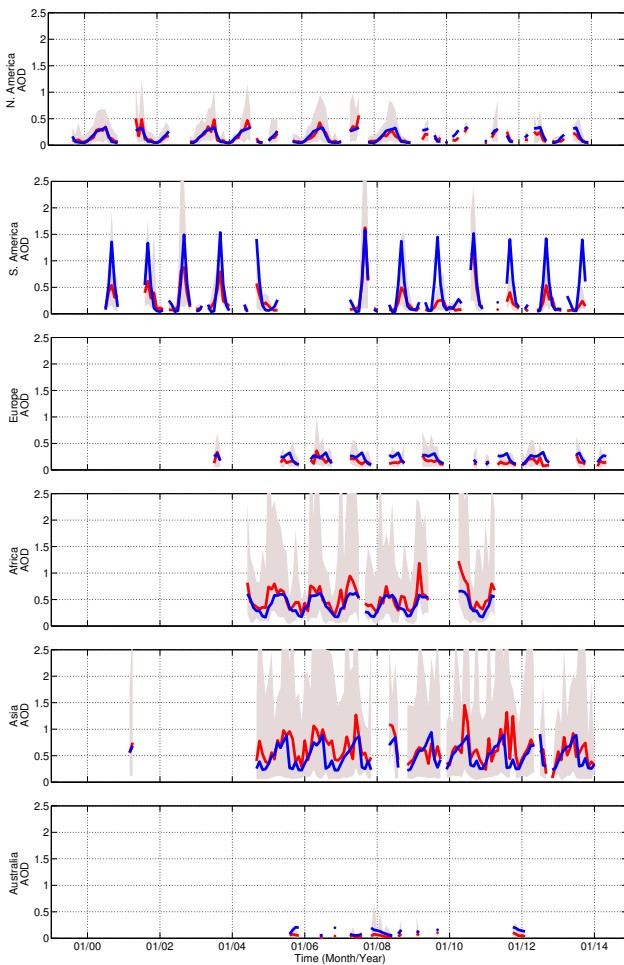

**Figure 20.** The monthly means of different data windows (North America, South America, Europa, Africa, Asia and Australia). The red colour indicates $\tau_{\text{AER}}$ values and the blue colour indicates the $\hat{\tau}$ values.





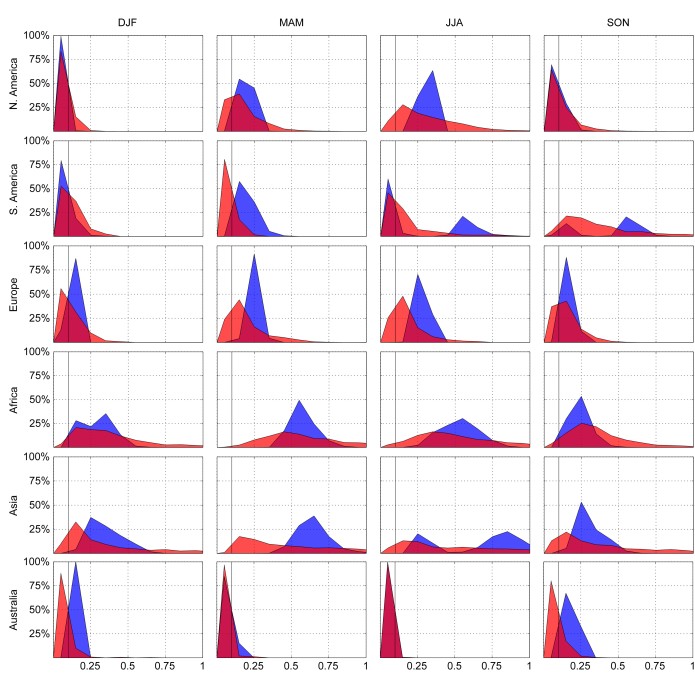

**Figure 21.** The normalized histograms of the seasonal AOD values from $\tau_{\mathrm{AER}}$ (red) and $\widehat{\tau}$ (blue) of different data windows.



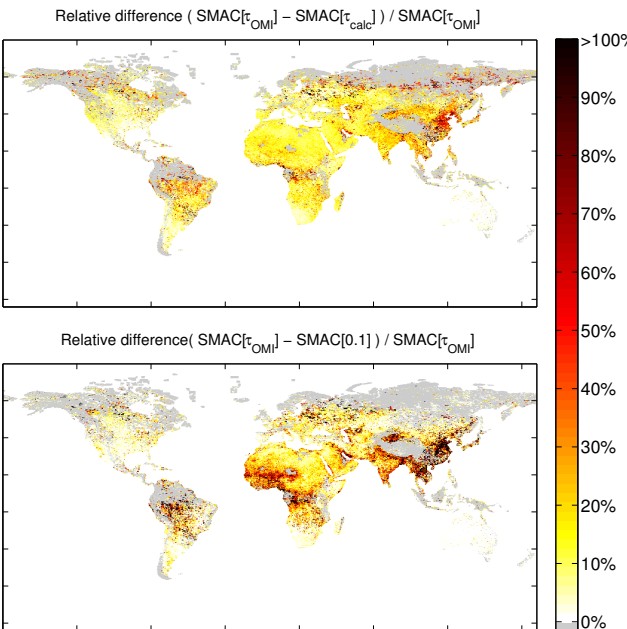

**Figure 22.** The medians of relative differences (magnitude values) of surface reflectance values calculated by SMAC in the year 2010. The differences between surface reflectance values calculated using $\tau_{OMI}$ and $\hat{\tau}$ are shown in the top figure, and the relative differences between surface values reflectance calculated using $\tau_{OMI}$ and the constant value 0.1 are shown in the bottom figure. The grey colour indicates the areas without data.



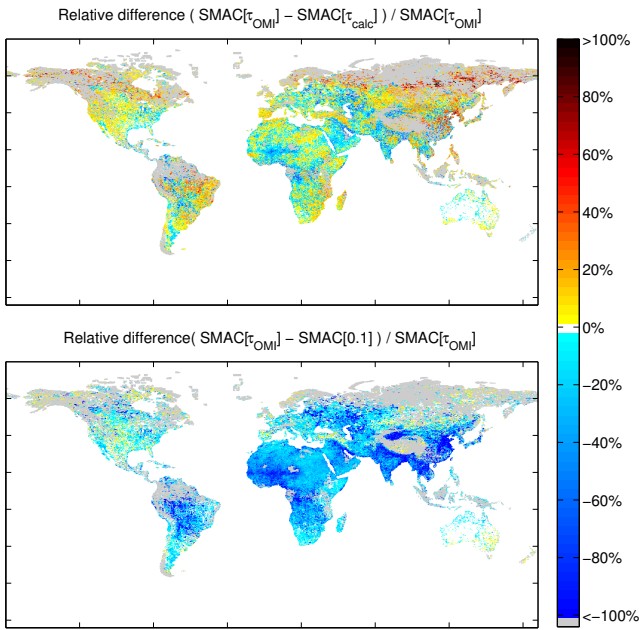

**Figure 23.** The medians of relative differences of surface reflectance values calculated by SMAC in the year 2010. The differences between surface reflectance values calculated using $\tau_{OMI}$ and $\hat{\tau}$ are shown in the top figure, and the relative differences between surface values reflectance calculated using $\tau_{OMI}$ and the constant value 0.1 are shown in the bottom figure. The grey colour indicates the areas without data.

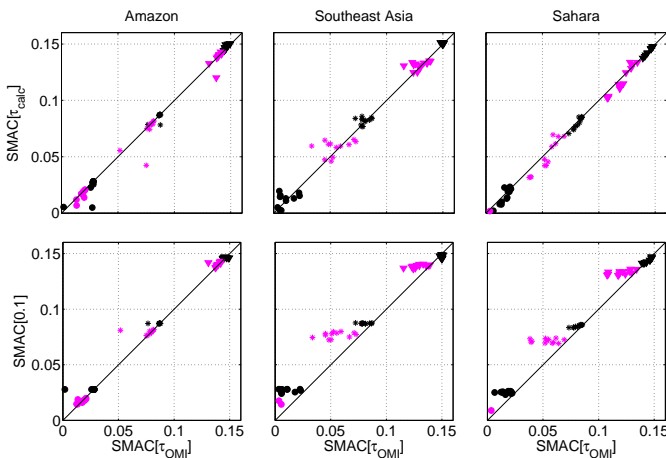

**Figure 24.** The surface reflectance values simulated by SMAC for subclasses 1, 24 and 39 (the Amazon and parts of Central America, Mainland Southeast Asia and the Sahara and the Middle East, respectively) for each month in the year 2006. The assumptions used for constant values for the satellite zenith angle are $0°$ (black) and $40°$ (magenta) and for ToA reflectance 0.05 (circle), 0.1 (asterisk) and 0.15 (triangle).



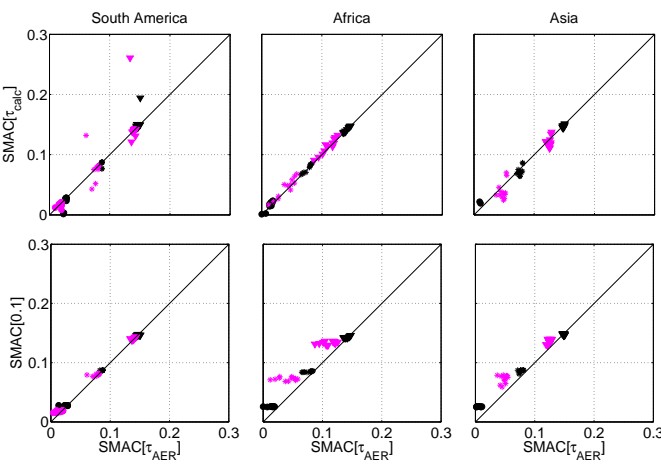

**Figure 25.** The surface reflectance values simulated by SMAC for three inspection windows (South America, Africa and Asia) for each month. The assumptions used for constant values for the satellite zenith angle are $0°$ (black) and $40°$ (magenta) and for ToA reflectance 0.05 (circle), 0.1 (asterisk) and 0.15 (triangle).