# Peer review of "An Aerosol Optical Depth time series 1982-2014 for atmospheric correction based on OMI and TOMS Aerosol Index"

_Atmospheric Measurement Techniques, 2016_

## Referee Comment (RC3)

Review of manuscript AMTD-180 by Jääskeläinen et al

**Summary**

This paper describes a procedure to create a climatological representation of aerosol optical depth over the continents for the period 1982-2014 using the OMI and TOMS aerosol index (AI), via an AI-to-AOD conversion procedure that involves the use of MODIS and OMI AOD satellite products.

The main purpose of the analysis is to replace a constant AOD value of 0.1 currently used in the CLARA-A2-SAL project with an estimate derived from the analysis in this paper, which, in the opinion of the authors, is a more realistic value.

**Review**

Although the authors have made an effort to develop a sound methodological approach, I do not believe they have achieved their proposed goal of deriving a realistic quantitative representation of the background atmospheric aerosol load over land which over most of the world is associated with sulfate-based industrial aerosols and biological particles, which is precisely what the AI is not.  Since the AI represents only a partial description of the global aerosol load, I do not believe their goal of deriving a realistic product is actually achievable.  For that reason, I do not think this work is publishable in its current form.  In the review below I offer a few recommendations mostly on the correct interpretation of the AI data to arrive at more realistic representation of the AI in terms of AOD only in the regions where the AI may indeed be used as proxy of most of the atmospheric aerosol load.

**Main comments**

My main criticism of this work is this work has to do with the over-interpretation of the AI as a proxy of the total column AOD, as well as its miss-interpretation in conditions where the residual quantity is associated with other non-aerosol related effects.   As it has been well documented the AI is only sensitive to elevated aerosol layers (about 2km and higher above the surface) of smoke, desert dust, and volcanic ash. Thus, the AI cannot be interpreted as a proxy of the total AOD column everywhere, and neither can it be interpreted as being representative of aerosol types other than optically thick layers of dust and smoke.  As the residual quantity it is, the AI is a representation of any wavelength dependent process unaccounted for by a simple radiative model representation of the Earth's atmosphere where molecular scattering and ozone absorption are the only radiative transfer processes explicitly included. Positive AI values larger than about 1.0 are generally associated with the absorption effects of layers of smoke, dust or volcanic ash located at least 2.0 km above the surface.  AI values less than 1.0 over land are undistinguishable from those associated with non-aerosol related effects such as wavelength dependent surface reflection effects (especially over the arid and semi-arid regions of the world) and scattering effects of clouds.  Thus, in the analysis carried out in this manuscript AI values lower than 1.0 should not be used.

Special care should be exercised to avoid anomalous positive AI values (often larger than unity) that are commonly observed at high latitudes in the late fall and winter seasons in both hemispheres. The nature of these anomalous AI values is not well understood, but it appears to be related to a breakdown at high solar and satellite zenith angles of the Lambertian approximation used in the AI calculation.

Based on the above stated considerations the AI signal can be considered a reasonable proxy of the total columns aerosol load only over regions where either dust, smoke or a dust-smoke combination account for most of columnar aerosol content yielding AI values larger than 1.0. Those regions include the well-known tropical and sub-tropical regions of Africa and both South and Central America where the AI signal is associated with the presence of optically thick smoke layers as well as the so-called dust belt that contains the world's major dust sources.

The description of the different data sets presented in Table 2 is confusing and misleading. The authors seem very unfamiliar with the AI data sets they are using. The TOMS v8 algorithm using the 331 and 360 nm channels is applied uniformly to both Nimbus-7 and Earth Probe observations. The earlier version (v7) made use of 340-380 nm for Nimbus7 and 331-360 nm for Earth Probe. According to the tabulated information the authors may actually be using v7 data for both sensors. The v8 data sets should be used. Otherwise, a scaling factor should be applied to the 331-360 AI which is about 25% lower than the 340-380 nm AI definition due to the wavelength separation.

Earth-Probe TOMS AI data after 2001 should not be used. A serious degradation issue affecting the sensor diffuser produces anomalously high AI values that must be ignored in any kind of trend analysis [Kiss et al., 2007].

**Other comments**

The reported wavelength-pair (342.5-388 nm) used for the calculation of the OMAERO AI parameter is at odds with the 354-388 nm pair reported in the literature [Torres et al., 2007].

It is not clear why the authors have chosen to work with the OMAERO AI. The obvious choice should be the OMTO3 AI product that uses the same wavelengths and the same algorithm as the TOMS V8 products. The V8 AI algorithm applied to Nimbus7 TOMS, Earth-Probe TOMS and Aura OMI (OMTO3) uses an algorithm that accounts for the presence of clouds at realistic location above the surface (MLER model). OMAERO AI uses a simple approximation (LER model), in which clouds are placed at surface level. These algorithmic differences produce significant AI difference in the presence of clouds and cloud-aerosol mixtures (Penning de Vries and Wagner, 2010).

The authors make use of MODIS and OMAERO AOD retrievals to transform the AI into the AOD space. More information on this procedure is needed. Which MODIS data is used? If the Dark Target MODIS (DTM) data is used, how do the authors handle the lack of DTM data over most of the world's arid and semi-arid areas? Please include key references to MODIS AOD validation studies. A justification for the use of the OMAERO product in this analysis should be provided. I am not aware of any comprehensive validation analysis of this product under different of aerosol conditions to support its application in a global product as intended in this analysis. Limited multi-sensor comparisons to AERONET observations

[Ahn et al., 2014; Carboni et al., 2014], shows significantly poorer OMAERO statistics relative to other satellite data sets.

The representativity of the resulting monthly long-term AOD record should be evaluated by comparison to other available multiyear records such as MODIS and MiSR (2000-present), SeaWIFS (1997-2010) and TOMS (1979-2001).

**References**

Ahn, C., O. Torres, and H. Jethva *(*2014*),* Assessment of OMI near-UV aerosol optical depth over land*,* J. Geophys. Res. Atmos.*,* 119*,* 2457–2473*, doi:*10.1002/2013JD020188*.*

Carboni, E., et al.: Desert dust satellite retrieval intercomparison, Atmos. Meas. Tech. Discuss., 5, 691-746, doi:10.5194/amtd-5-691-2012, 2012.

Kiss P., I.M. Janosi, and O. Torres, Early calibration problems detected in Earth Probe TOMS aerosol signal, *Geophys. Res. Letters, 34* L07803, doi: 10.1029/2006GL028108, 2007.

Penning de Vries, M. and Wagner, T.: Modelled and measured effects of clouds on UV Aerosol Indices on a local, regional, and global scale, Atmos. Chem. Phys., 11, 12715-12735, doi:10.5194/acp-11-12715-2011, 2011.

Torres, O., A. Tanskanen, B. Veihelman, C. Ahn, R. Braak, P. K. Bhartia, P. Veefkind, and P. Levelt, Aerosols and Surface UV Products from OMI Observations: An Overview, , *J. Geophys. Res.,* 112, D24S47, doi:10.1029/2007JD008809, 2007

---

## Referee Comment (RC1) · Anonymous Referee #1 · 1 Jun 2016

Atmospheric correction is needed to create land surface albedo data sets, and requires AOD. For the CLARA-A2-SAL albedo data set created from AVHRR measurements, for the time period 1982-2014, there is no available AOD product. However, UV aerosol index (AI) is available. This study presents a method to relate AI to AOD, and therefore provide an atmospheric correction for use in CLARA-A2-SAL processing. This is an improvement over the last version of the albedo data set, which used a constant AOD, which would therefore introduce some regional/seasonal biases into the albedo.

The research is important and the basic premise is sound. However I have some concerns about some aspects of the analysis, and some parts are unclear. My scientific comments are as follows:

[Figure]

MODIS AOD is used to filter the OMI AI. However, from Table 2, it seems like 3 different versions of the MODIS AOD data were used for different parts of the time period. This is somewhat surprising to me since the current version, Collection 006, has been available for about 3 years now, and the differences between the data versions have been documented to be large in some versions. So this would potentially introduce some discontinuities in the data set. I can't think of a good reason for using multiple versions of the MODIS data since all the data are available freely and it should not be too much of a burden to obtain the latest versions. I therefore strongly recommend that the analysis is repeated using consistently the latest data version (C006), rather than a mixture of this and older obsolete versions.

It is also not quite clear to me what time period of MODIS data are used. Section 3 suggests that only the time period 2005-2008 was used, but Table 2 gives different data versions for different time windows. This should be clarified. Whichever period is used, the latest MODIS data should be used.

On a related note, the authors don't say which MODIS AOD data product they are using (Dark Target, Deep Blue, or a combination). This should be stated. Both have advantages and limitations. For example Dark Target gives no coverage over deserts (Deep Blue does), while Dark Target has better coverage over tropical forests. Collection 006 contains a combined data set from both algorithms, which may be optimal here. Otherwise there will be lots of data coverage gaps. However in Figure 9, there is data over deserts, so perhaps Deep Blue is used. But it is not stated anywhere in the manuscript. And if not, then how is the AOD-AI regression done without MODIS AOD data over these regions, since section 3.4 says the regression is pixel-wise?

In section 3.1 it is not clear exactly how the OMI 550 nm AOD is created – specifically, the paper does not say where the Angstrom exponent is obtained from. It sounds like the AOD is estimated from each wavelength (with an Angstrom exponent from an unknown source), and then the estimates from each of the 5 wavelengths propagated to 550 nm are averaged. Is that right? Would a better way not be to use all 5 wave-

lengths together with the Angstrom power law to derive AOD at 550 nm and Angstrom exponent in a self-consistent way?

Section 3.3, I don't think that deseasonalisation of the AOD and AI data makes physical sense, and I am concerned that this will in fact introduce regional and seasonal artefacts into the data. AI depends on aerosol composition (amount and degree of absorption), altitude, and the underlying surface. This can vary widely from season to season within a given location. For example, patterns of biomass burning and dust aerosol tend to be highly seasonal. Vegetation phenology gives pronounced changes in the underlying surface cover, and seasonal differences in temperature and aerosol sources affect the aerosol height. All of these will modulate the AOD-AI relationship, and so there will be seasonally-dependent relationships in many regions. Yet as the authors note, the deseasonalisation step produces a seasonally-independent relationship. This will therefore introduce artefacts. For example, it could be responsible for some of the discrepancies in Figure 21.

Equation 5, what exactly are the 'modified' AOD and AI here? I did not find a definition for how these are different from the normal AOD and AI. Or does this refer to the deseasonalised data? This should be made clear.

Table 2 also shows that the wavelengths that AI is calculated from differ between Nimbus-7 TOMS, Earth Probe TOMS, and OMI. Since UV aerosol extinction and absorption exhibit spectral dependence, these wavelength differences mean that AI calculated for the same aerosol would differ between the sensors. This effect was not really discussed but should ideally be quantified.

The UV AI is also only sensitive to aerosols which are light-absorbing in the UV. So weakly-absorbing or nonabsorbing aerosols, such as sulphates, will have no AI signal. Yet they will contribute to the AOD and so affect the atmospheric correction. The regression may account for this, to an extent – I suppose it will contribute to the term beta in Equation 5. However if the loading of nonabsorbing aerosols is variable in time,

then this variation can't be captured by beta. The partition between nonabsorbing and absorbing aerosols in the AOD to AI conversion could be discussed in a bit more detail. Or is the pixel just completely discarded if the AI is too low, and not used to estimate albedo at all? This is hinted at in sections 2.2 and 3.1, but could be stated more explicitly.

The presented method, whereby AOD is estimated from AI, is clearly a better assumption than taking a constant value of AOD as was done in the first version of the albedo data set. However, in light of the above issues, I recommend that the manuscript is revised and re-reviewed after the above aspects have been clarified. Then it will be easier to understand the subtleties of what is done.

---

## Referee Comment (RC2) · Anonymous Referee #2 · 9 Jun 2016

The paper describes a method to produce a long-term aerosol optical depth (AOD) dataset reaching back to 1982. The purpose of this AOD dataset is its use for atmospheric correction of an AVHRR surface albedo time series 1982 - 2014. This is an important application and the AOD dataset is of high value, since no global AOD dataset suitable for this purpose exists over this long period. The title of the paper clearly states this specific limitation to one intended application of the AOD dataset. The final results for atmospheric correction prove the potential of the created dataset. However, the paper text is too short and needs to be extended to clearly describe the method used to produce the AOD dataset (e.g. snow / ice discrimination method, gap filling method, sub-class building, exclusion of low / high aerosol index, ...). Many

figures need to be described in the text: what the reader can see, what conclusion is drawn from them, what statement shall be highlight with it. In addition, the discussion of the impact of assumptions and the results achieved needs to be largely enhanced (e.g. fixed AOD over snow and ice, impact of differences between morning and afternoon orbits, ...). In particular the sensitivities of the AI to other parameters (foremost aerosol layer height, but also surface albedo, geometry, used UV wavelength pair) needs to be discussed. Additionally, the omission of non-absorbing aerosols as part of total AOD by the AI needs to be discussed. Furthermore, a conclusion section of only 7 lines is not suitable for a scientific paper. You need to summarize / discuss: impact of most critical assumptions, what have you achieved, what does a general reflectance increase mean, for which application is the mean validation on large aggregates sufficient, OMI AOD is not perfect - but taken here as truth... I therefore recommend a major revision of the paper. I recommend to start out from a discussion of the required accuracy for an AOD dataset to be used for atmospheric correction; this would then more clearly distinguish the atmospheric-correction AOD dataset from an AOD dataset for aerosol studies. In particular it should be stated which use of the AVHRR albedo dataset the authors have in mind (e.g. change detection of more qualitative and step-wise large differences over time, climate monitoring with small trends only to be detected in a noisy but stable time series), because this will determine the needed albedo accuracy and consequently the required AOD accuracy and stability. In the final discussion the achieved AOD accuracy can then be assessed in comparison to the assumption of a fixed AOD = 0.1. It needs to be discussed in how far the method does only correct for absorbing aerosols (excluding AI

poral scales would one expect to reproduce realistic aerosol variability, where one you expect to smoothen them?

Further comments: To make up for longer text, some of the figures are not necessary and can be deleted or combined. The authors should consider reducing figures: 1 (describing the main stability over long time but regional seasonal cycles in the text will suffice), 2 (one of the two maps is sufficient, aren't they adding up to 100%?), 3 (can be explained in text), 4 (better describe in text the principles for building the sub-classes), combine fig 8 and 10 into one flow chart with optional boxes; and tables: 5 (can be explained in 1 or 2 sentences in the text). The authors should make clearer in the title and text that they are discussing a time series of global maps (i.e. with regional AOD variability) to distinguish from a global averaged time series. This will then support the added value discussion of providing spatial information for the atmospheric correction. Spatial resolution of all datasets needs to be provided. English usage needs to be improved by involving a native English speaking person; e.g. articles are often missused, the word "manifolding" should be replaced (several times). Reword "TOMShomogenize" (p. 7 / I. 13). There are a number of vague statements which should be made more precise / quantitative; e.g. "sufficient" (p.1 / I.7), "long enough (p. 1 / I. 16), "a little bit too coarse (P. 3 / I. 23), "by a little" (p. 4 / I. 15), "some local inspections" (P. 5 / I. 15), "not so much" (p. 5 / I. 23), ...

Detailed comments: The last paragraph of section 1 (structure of the paper) should be shortened to only give one main heading for each section; further detail is not needed here. Section 2.1: EP-TOMS is not used and therefore needs not to be discussed at all. P. 3 / I. 17: MODIS AO is a retrieval, not an estimation (higher accuracy). P.3 / I. 21 -24: which land cover dataset do you use? Section 3.1: where do tau-UV and alpha come from? End of section 3.2 and later on: you mix up "areas" and "classes" – please be consistent to avoid confusing the reader. P. 5 / I. 15/16: I do not understand these statements – please explain what you mean. Section 3.2: this is very important to discuss the limitations / assumptions, but needs extension P. 5 /
I. 26: give minimum and maximum number of pixels; also I. 28 Fig. 6: better show results with AI \* cos (theta), since you use this quantity; also better colour bar should be used to show variability where most data points lie (e.g. between 0.5 and 0.8) P. 5 / I. 29/30: Correlations of 0.5 are still guite weak - I would thus be more cautious and rather conclude, that the method can only be used for parts of the dataset to construct reliable AOD P. 6 / I. 10: I do not understand why you need the ordering - isn't this just the weighted average? P. 6 / I. 13: a vector of what? P. 6 / I. 18: explain "after additional restrictions" P. 6 / I. 22: explain how you divide them P. 6 / I. 28/29: I do not understand this sentence; is the simplest also the best one or at least equally good as others? I suggest to show one example time series over those steps to illustrate better what you do; also a map of regression coefficients could be illustrative Start of section 4: motivate, why you need two different approaches P. 7 / I. 30: how exactly do you treat cases with AI outside the range [0.5, 4.5]? omit, set to 0.5 and 4.5, respectively, ... P. 7 / I. 29-31: why do you use two steps of spatial regridding? Fig. 9 needs discussion: many values too high (e.g. Scandinavia, California, Siberia, SouthEastAsia, Tibetan plateau, Himlaya, ...), mountains come out, compare to OMI AOD retrieval map

P. 8 / I. 15-18: I am not convinced why you use 3 years before and after the gap - motivate and explain P. 9 / I. 8: if the annual cycle was the same over all years, then you could produce one long-term climatology dataset, but there are intra-annual variations, one potential strength of your dataset P. 9 / I. 9: Tropic of Capricorn is the Southern - you want to point to the Northern (sub-) tropical maximum over the Sahara latitude? P. 9 / I. 12-22: this is not very clear (why should the more accurate MODIS dataset have less seasonality) P. 9 / I. 26-28: a difference of 0.3 is very large (given mean global AOD over land of  $\sim 0.2$ ); also next paragraph: you should talk of large differences, but say better, that they are still smaller than with assuming a fixed AOD=0.1 P. 10 / I. 3 onward: please state in how far the 3 example classes are representative for your analysis of all classes. Do they show best, worst or typical results? P. 10 / I. 32: please add AERONET reference: Holben, B.N.; Eck, T.F.; Slutsker, I.; Tanr'e, D.; Buis, J.P.; Setzer, A.; Vermote, E.; Reagan, J.A.; Kaufman, Y.J.;
Nakajima, T.; et al. AERONET—A federated instrument network and data archive for aerosol characterization. Remote Sens. Environ. 1998, 66, 1-16. p. 11 top: typical satellite AOD validation uses a window of 50x50 km2 for spatial matching; you need to discuss whether you are not creating artificial variability on pixel level P. 11 / I. 10 onward and fig. 10+11: use more specific names, not the continents, where the small test regions lie in - this is misleading Fig. 22: better show absolute differences, not relative – otherwise you highlight larger relative errors over dark surfaces P. 12 / I. 14: can you draw a quantitative conclusion rather than saying that reflectances tend to be higher? Fig. 11: why do you not make a scatter plot of AODs? Add discussion in the text: El Nino Indonesia fires can be seen in 1997, lat 60N much too high, Sahara under-estimated/ biomass burning over-estimated, ... Fig. 12: why are there several curves for each category? Fig. 13: global mean AOD over land is  $\sim$ 0.2 - so you cannot make it that crude - you have extreme differences + and - 0.7 or so; better show the range -0.25 to 0.25 and exclude the other regions Fig. 18: you show partly very large differences: peaks, distribution shapes, double peaks; how can AOD be >1 with your method? Use a better-suited x-axis (e.g. 0-1) Fig. 19 / text: discuss whether those 6 regions are suited to grasp all global variability of aerosol and surface conditions Fig. 21: state in text partly significantly wrong seasonality (thus limiting the capabilities for atmospheric correction to use for assessing seasonal changes) Fig. 22-24: which wavelength or band reflecances? Fig. 22: figure title should be "relative difference of corrected reflectance values" ("magnitude values" is inappropriate terminology); better show scatter plots; I would prefer to see absolute values of reflectance differences; use better colour bar: large areas go from pale yellow to dark yellow (become worse, hard to be seen), some areas become better (from dark red to pale red); I would distinguish negative and positive values Fig. 24: why not again year 2010?

---

## Author Comment (AC2) · 17 Nov 2016

Author's response to Interactive comments of Anonymous Referee #2 on "An Aerosol Optical Depth time series 1982–2014 for atmospheric correction based on OMI and TOMS Aerosol Index" by E. Jääskeläinen et al.

We thank the referee for careful reading of our manuscript and for the detailed comments. We will incorporate these comments to the revised manuscript. Below, we list referees' comments followed by our answers (in blue). The pages and lines included in our answers refer to the revised manuscript.

The paper describes a method to produce a long-term aerosol optical depth (AOD) dataset reaching back to 1982. The purpose of this AOD dataset is its use for atmospheric correction of an AVHRR surface albedo time series 1982 - 2014. This is an important application and the AOD dataset is of high value, since no global AOD dataset suitable for this purpose exists over this long period. The title of the paper clearly states this specific limitation to one intended application of the AOD dataset. The final results for atmospheric correction prove the potential of the created dataset. However, the paper text is too short and needs to be extended to clearly describe the method used to produce the AOD dataset (e.g. snow / ice discrimination method, gap filling method, sub-class building, exclusion of low / high aerosol index, : : : ).

Many figures need to be described in the text: what the reader can see, what conclusion is drawn from them, what statement shall be highlight with it.

There are now more descriptions of the figures (Figures 3, 4, 7, 8, 11, 18-22 ).

In addition, the discussion of the impact of assumptions and the results achieved needs to be largely enhanced (e.g. fixed AOD over snow and ice, impact of differences between morning and afternoon orbits, : : : ).

The AOD time series presented in the manuscript is the first version of it, so some of the most difficult aspects (like AOD values over ice and snow or the impact of the different orbits) are left to be solved in the future. The atmosphere is thin over the poles so the accurate magnitude of the atmospheric correction is not that critical. Hence, we don't currently calculate AOD over the permanently snow covered areas. Because we cannot define the AOD values precisely over the AVHRR orbit to be used in the surface albedo retrieval, the first step to that direction is the daily value.

In particular the sensitivities of the AI to other parameters (foremost aerosol layer height, but also surface albedo, geometry, used UV wavelength pair) needs to be discussed.

We added more text about the sensitivities of the AI to other parameters to the manuscript, P. 5/L. 29-31.

Additionally, the omission of non-absorbing aerosols as part of total AOD by the AI needs to be discussed.

Even though the negative AI values are discarded, the regressions are made by using the total AOD from the OMI instrument together with the land use classification information. So the use of the absorbing AI values still produce AOD values which are more close to the total AOD than to only absorbing AOD values. Text added to the manuscript, P. 4/L. 8-10, P. 10/L. 27-30.

Furthermore, a conclusion section of only 7 lines is not suitable for a scientific paper. You need to summarize / discuss: impact of most critical assumptions, what have you achieved, what does a general reflectance increase mean, for which application is the mean validation on large aggregates sufficient, OMI AOD is not perfect - but taken here as truth...

The conclusions section is now expanded a little bit and a new section, Section 7: Discussion and conclusions is added to the manuscript; both changes will now address the mentioned matters in more detail.

I therefore recommend a major revision of the paper.

I recommend to start out from a discussion of the required accuracy for an AOD dataset to be used for atmospheric correction; this would then more clearly distinguish the atmospheric-correction AOD dataset from an AOD dataset for aerosol studies. In particular it should be stated which use of the AVHRR albedo dataset the authors have in mind (e.g. change detection of more qualitative and step-wise large differences over time, climate monitoring with small trends only to be detected in a noisy but stable time series), because this will determine the needed albedo accuracy and consequently the required AOD accuracy and stability. In the final discussion the achieved AOD accuracy can then be assessed in comparison to the assumption of a fixed AOD = 0.1.

The minimum requirement was to achieve a better accuracy by using the constructed AOD time series compared to the constant AOD 0.1. Because the CLARA-A2 SAL data are used for climate models, it is more important that the overall quality and levels of the data set are right than that all the dynamic changes are detected. For the next versions of the AOD time series, this matter will be taken into account, a note of this is added to the new section in the manuscript, section 7 titled as Discussion and conclusions.

It needs to be discussed in how far the method does only correct for absorbing aerosols (excluding AI < 0) and how this will affect the AOD and albedo values. Also the impact of the difference between total AOD from MODIS and absorbing-aerosol AOD from AI in the regression of the method needs discussion.

This matter is now assessed in the manuscript, P. 4/L. 8-10, P. 10/L. 27-30. Even though the AOD values from the constructed AOD time series are calculated by using the absorbing AI values, they are not exactly absorbing AOD values, because the regressions are derived for AI and total AOD. For this reason, it is not so problematic to compare the constructed AOD time series to the total AOD from MODIS.

The evaluation is too much done with global / zonal and long-term averages – the added value of the AOD daily maps lies in the spatial and temporal patterns for the atmospheric correction. Also, providing those daily maps contains the risk of introducing additional noise into the datasets – this needs to be assessed, at least with exemplary studies.

The evaluations could have been done more from daily values, that's true. Below there are some example figures. In the first three figures, the AOD data from the seven AERONET stations (red) are compared to the constructed AOD time series data (blue). The data looks very similar with the monthly mean figure (Fig. 17 in the manuscript). The fourth figure is similar as the first figure, but there are data only from one year, 2006, to show better the daily values. The constructed AOD values have clearly less variability than the in situ AOD values.

The CLARA-A2 SAL product is reported in pentad and monthly means. The AOD daily maps may introduce additional noise, but the pentad and the monthly means of albedo product will filter some of the noise away. We added a note of this to the new section in the manuscript, section 7 titled as Discussion and conclusions.

[Figure]

[Figure]

[Figure]

On what spatial and temporal scales would one expect to reproduce realistic aerosol variability, where one you expect to smoothen them?

The daily values from three AERONET stations in Africa over one year, 2006, are compared to the constructed AOD values from the same locations, and the results are shown in the figure above. The constructed AOD have clearly less variability than the in situ AOD values, but the constructed AOD values are not smooth over the year. The variability in the constructed AOD values is not as distinct as it is in the in situ AOD values, but it is still faithful to the main features of the in situ AOD.

Further comments: To make up for longer text, some of the figures are not necessary and can be deleted or combined. The authors should consider reducing figures: 1 (describing the main stability over long time but regional seasonal cycles in the text will suffice), 2 (one of the two maps is sufficient, aren't they adding up to 100% ?), 3 (can be explained in text), 4 (better describe in text the principles for building the sub-classes), combine fig 8 and 10 into one flow chart with optional boxes; and tables: 5 (can be explained in 1 or 2 sentences in the text).

Figure 1: Removed and the content are now explained in the text, P. 4/L. 2-5
Figure 2: The positive values map is removed and the text edited accordingly, P. 4/L. 5-8.
Figure 3: Removed and text added, P. 4/L. 24.
Figure 4: Because the division to the subclasses is made somewhat subjectively, the figure makes it easier to grasp that how the division is done.
Figures 8 and 10 combined.
Table 5: Removed the table and the contents are now explained in the text, P. 13/L. 26-28, P. 14/L. 6-7.

The authors should make clearer in the title and text that they are discussing a time series of global maps (i.e. with regional AOD variability) to distinguish from a global averaged time series. This will then support the added value discussion of providing spatial information for the atmospheric correction.

Mention of this matter is added to the text, P. 2/L. 13, and we modified the title of the manuscript.

Spatial resolution of all datasets needs to be provided.

The spatial resolutions of the TOMS, OMI and MODIS data sets are collected in the Table 2, and the resolutions of the land use data are added to the text, P. 3/L. 23-24

English usage needs to be improved by involving a native English speaking person; e.g. articles are often miss-used, the word "manifolding" should be replaced (several times).

All detailed suggestions of the reviewers have been carried out. However, it is not easy to satisfy all reviewers/readers simultaneously, because the use of English language has so large variation. Namely, the language of the original manuscript had been officially checked by a native English speaker who is doing language checking professionally. We shall give the feedback of the reviewers to the language checking company.

Reword "TOMS- homogenize" (p. 7 / l. 13).

We changed the word "TOMS-homogenize", P. 7/L. 28-29, P. 8/L. 1, 17, 19

There are a number of vague statements which should be made more precise / quantitative; e.g. "sufficient" (p.1 / l.7), "long enough (p. 1 / l.16), "a little bit too coarse (P. 3 / l. 23), "by a little" (p. 4 / l. 15), "some local inspections" (P. 5 / l. 15), "not so much" (p. 5 / l. 23), : : :

These statements are made more precise.

Detailed comments: The last paragraph of section 1 (structure of the paper) should be shortened to only give one main heading for each section; further detail is not needed here.

The paragraph is shortened as requested.

Section 2.1: EP-TOMS is not used and therefore needs not to be discussed at all.

The AI data from EP-TOMS is used for the time period 1996-2001, Subsection 2.1: P. 3/L. 4, Subsection 4.2.

P. 3 / l. 17: MODIS AOD is a retrieval, not an estimation (higher accuracy).

The word is changed accordingly in the manuscript.

P.3 / I . 21 -24: which land cover dataset do you use?

As mentioned on P. 3/L. 21-25, the used land cover data sets are the AVHRR Land Use Classification, where the spatial resolution is 1 degree, and Global Land Cover 2000.

Section 3.1: where do tau-UV and alpha come from?

The tau-UV comes from the OMAEROe product, AOD at [342.5nm, 388nm, 442nm, 463nm, 483.5nm] and the alpha is calculated from these values. The Ångström exponent is calculated once from each wavelength pair (ten combinations altogether) and these exponents are then used to estimate the AOD at 550 nm by using the AOD values at suitable wavelengths (again, ten AOD values at 550 nm altogether). Text updated P. 5/L. 6-9

End of section 3.2 and later on: you mix up "areas" and "classes" – please be consistent to avoid confusing the reader.

The word "area" is now replaced by the word "subclass" whenever necessary.

P. 5 / l. 15/16: I do not understand these statements – please explain what you mean.

The subclass division is used for helping for example in deseasonalization and when determining the regression coefficients.  Some land cover types (such as deserts) are related to certain aerosols. Text edited, P. 5/L. 26

Section 3.2: this is very important to discuss the limitations / assumptions, but needs extension

We added more text of the sensitivities of the AI to other parameters to the manuscript, P. 5/L. 31-32

P. 5 /l. 26: give minimum and maximum number of pixels;
Added as requested, P. 6 /L. 7-8.

also l. 28 Fig. 6: better show results with AI * cos (theta), since you use this quantity; also better colour bar should be used to show variability where most data points lie (e.g. between 0.5 and 0.8)

The figure in question changed along with a better colour bar as requested, and the text updated as well (P. 6 /L. 9-10).

P. 5/ l. 29/30: Correlations of 0.5 are still quite weak – I would thus be more cautious and rather conclude, that the method can only be used for parts of the dataset to construct reliable AOD

It is true, the correlation of 0.5 is weak. The text is edited (P. 6/L. 12-14) as requested by concluding that the method provides probably more accurate AOD values in certain areas of the globe.

P. 6 / l. 10: I do not understand why you need the ordering – isn't this just the weighted average?

The ordering indicates the process how the weights are obtained. Text clarified, P. 6 / L. 26.

P. 6 / l. 13: a vector of what?

A vector where each value is added multiple times. Text edited, P. 6 / L. 28.

P. 6 / l. 18: explain "after additional restrictions"

Text edited P. 7 / L. 2, additional restrictions referenced to restrictions of AOD < 1 and SZA < 70.

P. 6 / l. 22: explain how you divide them

Word "divided" changed to the word "processed." P. 7/ L. 6

P. 6 / l. 28/29: I do not understand this sentence; is the simplest also the best one or at least equally good as others?

The simplest one was the best model of those which were equally good. Text edited P. 7/L. 12-13

I suggest to show one example time series over those steps to illustrate better what you do; also a map of regression coefficients could be illustrative

Below are the maps of chosen regression coefficients.

Chosen regression coefficients: AI

[Figure]

Chosen regression coefficients: constant

[Figure]

Start of section 4: motivate, why you need two different approaches

Text added P. 8/L. 8-10

P. 7 / l. 30: how exactly do you treat cases with AI outside the range [0.5, 4.5]? omit, set to 0.5 and 4.5, respectively, : : :

The AI outside the range [0.5, 4.5] are omitted, text added P.8/L. 16

P. 7 / l. 29-31: why do you use two steps of spatial regridding?

The data from the OMI instrument have different resolution that the data from the TOMS instrument (Table 2). The The data are homogenized by using the resolution of TOMS and it is done, because we want to avoid the difference in the data when using the AI data from two different data sets. The second spatial regridding is done to change the data back to the original resolution which is the resolution of CLARA-A2-SAL. Text added, P.8 / L. 15-18

Fig. 9 needs discussion: many values too high (e.g. Scandinavia, California, Siberia, SouthEastAsia, Tibetan plateau, Himlaya, : : : ), mountains come out, compare to OMI AOD retrieval map

We added more details of about the figure, P.8 / L. 28-32, P.9 / L. 1-4, and added an additional figure of about the absolute differences between the constructed AOD and MODIS-AOD. We compared the constructed AOD from the May 2005 to the MODIS-AOD from the same month and year. The accuracy requirement is not included for the mountains in CLARA-A2 SAL product, so it doesn't produce a problem if the AOD time series has difficulties in those areas.

P. 8 / l. 15-18: I am not convinced why you use 3 years before and after the gap
– motivate and explain

One year is not enough to catch the variability of the AI, so three years are a safer solution. This gap filling method is the most simple and robust solution for the time being. Text added P. 9/L. 13

P. 9 / l. 8: if the annual cycle was the same overall years, then you could produce one long-term climatology dataset, but there are intra-annual variations, one potential strength of your dataset

This is true, text edited P. 10/L. 8-10.

P. 9 / l. 9: Tropic of Capricorn is the Southern – you want to point to the Northern (sub-) tropical maximum over the Sahara latitude?

No, Tropic of Capricorn is right. The sentence was unclear, the highest AOD are due to the high AOD values from the Amazon area in September. Text is modified to be more clear, P. 10/L. 11.

P. 9 / l. 12-22: this is not very clear (why should the more accurate MODIS dataset have less seasonality)

MODIS-AOD have less seasonality in a sense of global monthly means. The standard deviations vary more (the grey area behind the lines). Text modified P. 10/L. 22-23.

P. 9 / l. 26-28: a difference of 0.3 is very large (given mean global AOD over land of 0.2); also next paragraph: you should talk of large differences, but say better, that they are still smaller than with assuming a fixed AOD=0.1

Yes, it is true that the difference of 0.3 is quite large. We modified Figure 11 by adding the absolute differences of zonal monthly means between OMI-AOD and the constant AOD value 0.1, where one can easily see, that the differences are smaller between OMI-AOD and the constructed AOD

time series compared to the differences between OMI-AOD and the constant value. Also, text edited accordingly, P. 11/L. 6-11.

P. 10 / l. 3 onward: please state in how far the 3 example classes are representative for your analysis of all classes. Do they show best, worst or typical results?

The results in Amazon subclass are weaker as expected compared to the other two subclasses, especially in the season SON, because the OMI-AOD vary a lot, from 0.2 to over unity and the linear regression cannot predict that well. In the subclass covering the Sahara and the Middle East, the aerosols are typically dust and there the linear regression using AI provides more accurate AOD. The Mainland Southeast Asia subclass is something between Sahara and Amazon. It is more typical subclass compared to all the others and the results are hence more typical. Text added as requested, P. 12 /L. 11-17.

P. 10 / l. 32: please add AERONET reference: Holben, B.N.; Eck, T.F.; Slutsker, I.; Tanré, D.; Buis, J.P.; Setzer, A.; Vermote, E.; Reagan, J.A.; Kaufman, Y.J.; Nakajima, T.; et al. AERONET—A federated instrument network and data archive for aerosol characterization. Remote Sens. Environ. 1998, 66, 1–16.

We added the reference, P. 12/L. 20.

p. 11 top: typical satellite AOD validation uses a window of 50x50 km2 for spatial matching; you need to discuss whether you are not creating artificial variability on pixel level

The authors did not understand this point. In this section we discuss the problem of comparing point wise in situ measurements and large satellite pixels, which we solve by comparing a distribution of several in situ measurements within the large pixel of the satellite. We are not doing any processing in this section that could cause pixel level variation.

P. 11 / l. 10 onward and fig. 10+11: use more specific names, not the continents, where the small test regions lie in - this is misleading

The names are changed to more specific ones, i.e. P. 12/L. 32-35, P. 13/L 1-2.

Fig. 22: better show absolute differences, not relative – otherwise you highlight larger relative errors over dark surfaces

Figure updated as requested, also text changed, P. 13/L. 22-23, 28-32.

Fig. 22: figure title should be "relative difference of corrected reflectance values" ("magnitude values" is inappropriate terminology); better show scatter plots; I would prefer to see absolute values of reflectance differences; use better colour bar: large areas go from pale yellow to dark yellow (become worse, hard to be seen), some areas become better (from dark red to pale red); I would distinguish negative and positive values

Figure updated as requested (from relative differences to absolute differences), also text changed, P. 13/L. 22-23, 28-32. We decided not to do scatter plots for the manuscript, because it won't show

where the differences are spatially. Negative and positive absolute differences are shown in Figure 20. The scatter plots of the means of the simulated SMAC values are shown below.

[Figure]

P. 12 / l. 14: can you draw a quantitative conclusion rather than saying that reflectances tend to be higher?

The quantitative conclusion added, P. 14/L. 22.

Fig. 11: why do you not make a scatter plot of AODs?

The point of that figure is to show how homogeneous the constructed AOD time series is.

Add discussion in the text: El Nino Indonesia fires can be seen in 1997, lat 60N much too high, Sahara under-estimated/ biomass burning over-estimated, : : :

We added more discussion about the zonal mean figure (Figure 8 in the revised manuscript) to the text, P. 10/L. 9-15.

Fig. 12: why are there several curves for each category?

There is a curve for each year (2005-2014), added the year information to the figure caption.

Fig. 13: global mean AOD over land is 0.2 - so you cannot make it that crude - you have extreme differences + and - 0.7 or so; better show the range -0,25 to 0.25 and exclude the other regions

The point of the figure is to show where the large positive AOD values occur in the calculated AOD time series in relation to MODIS-AOD. Below is the same inspection, but now only the differences from the range [-0.25, 0.25] are shown.

[Figure]

Fig. 18: you show partly very large differences: peaks, distribution shapes, double peaks; how can AOD be >1 with your method? Use a better-suited x-axis (e.g. 0-1)

The x-axis of Figure 18 is updated as requested. In the regressions the AI and AOD data were limited to the values where AOD < 1, but in the calculation of the daily AOD maps from the AI data there were no such limits.

Fig. 19 / text: discuss whether those 6 regions are suited to grasp all global variability of aerosol and surface conditions

The chosen regions do not cover all the possible aerosol scenarios, but they offer enough variability, that we can assess the use of the constructed AOD time series in the atmospheric correction. Added text to the manuscript, P. 13/L. 2-7.

Fig. 21: state in text partly significantly wrong seasonality (thus limiting the capabilities for atmospheric correction to use for assessing seasonal changes)

Text changed accordingly, P. 13/L. 13-17.

Fig. 22-24: which wavelength or band refletances?

The used wavelength band was 0.725-1.000 µm, text added: P. 13/L. 19-20.

Fig. 24: why not again year 2010?

That's true, it should have been year 2010. The figure in question is now updated by using the data from the year 2010.

---

## Author Comment (AC3) · 17 Nov 2016

Author's response to Interactive comments of Anonymous Referee #3 on "An Aerosol Optical Depth time series 1982–2014 for atmospheric correction based on OMI and TOMS Aerosol Index" by E. Jääskeläinen et al.

We thank the referee for careful reading of our manuscript and for the helpful comments. We will incorporate these comments to the revised manuscript. Below, we list referees' comments followed by our answers (in blue). The pages and lines included in our answers refer to the revised manuscript.

Summary
This paper describes a procedure to create a climatological representation of aerosol optical depth over the continents for the period 1982-2014 using the OMI and TOMS aerosol index (AI), via an AI-to-AOD conversion procedure that involves the use of MODIS and OMI AOD satellite products. The main purpose of the analysis is to replace a constant AOD value of 0.1 currently used in the CLARA-A2-SAL project with an estimate derived from the analysis in this paper, which, in the opinion of the authors, is a more realistic value.

Review
Although the authors have made an effort to develop a sound methodological approach, I do not believe they have achieved their proposed goal of deriving a realistic quantitative representation of the background atmospheric aerosol load over land which over most of the world is associated with sulfate-based industrial aerosols and biological particles, which is precisely what the AI is not. Since the AI represents only a partial description of the global aerosol load, I do not believe their goal of deriving a realistic product is actually achievable. For that reason, I do not think this work is publishable in its current form. In the review below I offer a few recommendations mostly on the correct interpretation of the AI data to arrive at more realistic representation of the AI in terms of AOD only in the regions where the AI may indeed be used as proxy of most of the atmospheric aerosol load.

Main comments
My main criticism of this work is this work has to do with the over-interpretation of the AI as a proxy of the total column AOD, as well as its miss-interpretation in conditions where the residual quantity is associated with other non-aerosol related effects. As it has been well documented the AI is only sensitive to elevated aerosol layers (about 2km and higher above the surface) of smoke, desert dust, and volcanic ash. Thus, the AI cannot be interpreted as a proxy of the total AOD column everywhere, and neither can it be interpreted as being representative of aerosol types other than optically thick layers of dust and smoke. As the residual quantity it is, the AI is a representation of any wavelength dependent process unaccounted for by a simple radiative model representation of the Earth's atmosphere where molecular scattering and ozone absorption are the only radiative transfer processes explicitly included. Positive AI values larger than about 1.0 are generally associated with the absorption effects of layers of smoke, dust or volcanic ash located at least 2.0 km above the surface. AI values less than 1.0 over land are undistinguishable from those associated with non-aerosol related effects such as wavelength dependent surface reflection effects (especially over the arid and semi-arid regions of the world) and scattering effects of clouds. Thus, in the analysis carried out in this manuscript AI values lower than 1.0 should not be used.

We were aware of this issue (the over-interpretation of the AI as a proxy of the total column AOD) when constructing the AOD time series. The motivation for the AOD time series is the need for daily AOD information for the years 1982-2014. As there is no such data set available, we have to construct it to avoid using a climatological solution in varying climate. The only existing homogenous aerosol related data set for the whole needed time period, from which to calculate this AOD time series, is AI data from TOMS and OMI instruments. It is true, that there are issues in using the AI data as a proxy for AOD information and the issues should have been described better in the manuscript. There is now a remark of this issue in the manuscript, P. 2/ L. 23-27, and also information of where the relation between AI and AOD is most reliable and where it is not. We have also now clarified in the manuscript (i.e. P. 4/L. 8-10 and P. 10/L. 28-30), that the use of the land use classification information in the calculations provides some information about the nonabsorbing aerosols.

The demarcation of limits of the data used is always difficult, and we decided to use AI values below 1.0 even though it produces possible problems by weakening the overall quality. For us, the atmospheric correction is the main interest, and the results obtained by using the constructed AOD as aerosol information in the atmospheric correction algorithm are quite promising. The use of AI values below 1.0 is also a conscious risk, but we will take this matter into account in the future versions of the AOD time series.

Estimating AOD of scattering aerosols at 550 nm using the UV information is justified, because the wavelength difference is relatively small to allow using the Ångström exponent relation. For absorbing aerosols one cannot expect to get a direct relationship between two wavelengths. In that case the statistical relationship between the amounts of scattering and absorbing aerosols seems to be typically strong enough to provide good enough results.

Special care should be exercised to avoid anomalous positive AI values (often larger than unity) that are commonly observed at high latitudes in the late fall and winter seasons in both hemispheres. The nature of these anomalous AI values is not well understood, but it appears to be related to a breakdown at high solar and satellite zenith angles of the Lambertian approximation used in the AI calculation.

The Solar zenith angle is over 70° at high latitudes in both hemispheres in the late fall and winter seasons, and the CLARA-A2 SAL product is not calculated over those high SZA values. So these anomalous AI values do not affect calculation of the AOD time series in question

Based on the above stated considerations the AI signal can be considered a reasonable proxy of the total columns aerosol load only over regions where either dust, smoke or a dust-smoke combination account for most of columnar aerosol content yielding AI values larger than 1.0. Those regions include the well-known tropical and sub-tropical regions of Africa and both South and Central America where the AI signal is associated with the presence of optically thick smoke layers as well as the so-called dust belt that contains the world's major dust sources.

We added and improved text to emphasize, that the constructed AOD works well in certain areas (i.e. Sahara), but might fail in others (i.e. Amazon): P. 2/L. 26-27, P. 6/L. 12-14, P. 12/L. 11-18..

The description of the different data sets presented in Table 2 is confusing and misleading. The authors seem very unfamiliar with the AI data sets they are using. The TOMS v8 algorithm using the 331 and 360 nm channels is applied uniformly to both Nimbus-7 and Earth Probe observations. The earlier version (v7)made use of 340-380 nm for Nimbus7 and 331-360 nm for Earth Probe.

According to the tabulated information the authors may actually be using v7 data for both sensors. The v8 data sets should be used. Otherwise, a scaling factor should be applied to the 331-360 AI which is about 25% lower than the 340-380 nm AI definition due to the wavelength separation.

We apologize, there were error in Table 2, the used version was indeed v8 and the wavelength range for AI calculation in Nimbus-7 is also 331-360 nm, the same as it is in Earth Probe. It is now corrected in the manuscript Table 2.

Earth-Probe TOMS AI data after 2001 should not be used. A serious degradation issue affecting the sensor diffuser produces anomalously high AI values that must be ignored in any kind of trend analysis [Kiss et al., 2007].

Yes, the TOMS-AI data 2002 onwards are omitted from the calculations, P. 9/L. 14-17.

Other comments
The reported wavelength-pair (342.5-388 nm) used for the calculation of the OMAERO
AI parameter is at odds with the 354-388 nm pair reported in the literature [Torres et al., 2007].

We apologize, there was an error in Table 2, the wavelength range for the calculation of the OMI-AI is indeed 354-388nm. It is now corrected in the manuscript Table 2.

It is not clear why the authors have chosen to work with the OMAERO AI. The obvious choice should be the OMTO3 AI product that uses the same wavelengths and the same algorithm as the TOMS V8 products. The V8 AI algorithm applied to Nimbus7 TOMS, Earth-Probe TOMS and Aura OMI (OMTO3) uses an algorithm that accounts for the presence of clouds at realistic location above the surface (MLER model). OMAERO AI uses a simple approximation (LER model), in which clouds are placed at surface level. These algorithmic differences produce significant AI difference in the presence of clouds and cloud-aerosol mixtures (Penning de Vries and Wagner, 2010).

Yes, this is true, we should have used OMTO3 data. In the future versions we will use the OMTO3 data. Note of this is in the new section in the manuscript, section 7 titled as Discussion and conclusions. See next answer for more information.

The authors make use of MODIS and OMAERO AOD retrievals to transform the AI into the AOD space. More information on this procedure is needed. Which MODIS data is used? If the Dark Target MODIS (DTM) data is used, how do the authors handle the lack of DTM data over most of the world's arid and semi-arid areas? Please include key references to MODIS AOD validation studies. A justification for the use of the OMAERO product in this analysis should be provided. I am not aware of any comprehensive validation analysis of this product under different of aerosol conditions to support its application in a global product as intended in this analysis. Limited multi-sensor comparisons to AERONET observations [Ahn et al., 2014; Carboni et al., 2014], shows significantly poorer OMAERO statistics relative to other satellite data sets.

The AOD data from Dark Target and Deed Blue aerosol retrieval algorithms are used. This detail is now clarified in the manuscript P. 5/L. 12-13, and we also added the references to the MODIS-AOD validation studies, P. 5/L. 12-13.

The OMAEROe AOD, after calculating to the AOD at 550 nm, are screened with MODIS-AOD (i.e. P. 4/L. 18-19, P. 5/L. 10-13,), so the used AOD are close to the MODIS-AOD values. It is true that we should have used OMTO3 data (see answer above). Below are two figures, where the surface reflectance values calculated by atmospheric correction algorithm SMAC are shown. The used assumptions for constant values for the satellite zenith angle are 0° (light blue, light red, light green, cyan, grey colours) and 40° (dark blue, dark red, dark green, purple, grey colours) and for ToA reflectance 0.05 (circle), 0.1 (asterisk) and 0.15 (triangle). The satellite azimuth angle is a constant 260°, water vapour content 2.5, integrated ozone 0.35 and pressure 1013. The Solar zenith angles, Solar azimuth angles and AOD at 550 nm for each month are chosen to be the most probable value from each subclass of the year 2010 from constructed AOD time series (blue), OMI-AOD (red), MODIS-AOD, SeaWIFS-AOD (cyan and purple) and MISR-AOD data (grey and black). The marker types and colours are used only in the first figure. The individual simulations are in the first figure, and the combined results are in the second one, where the dark blue indicates the surface reflectance values calculated by using constructed AOD as aerosol information and the light blue those values which were calculated by using the SeaWIFS-AOD and MISR-AOD values. The surface reflectance values calculated by using the constructed AOD, OMI-AOD, the constant value 0.1, SeaWIFS-AOD and MISR-AOD are compared to the surface reflectance values calculated by using MODIS-AOD values. From those results can be seen that even though the constructed AOD have some difficulties in the Amazon area, the values it produces for the surface reflectance by applying the SMAC algorithm do not essentially differ from the others. So the use of OMAERO-AOD as the basis data do not seems to cause major problems. Note of this is in the new section in the manuscript, section 7 titled as Discussion and conclusions.

[Figure]

[Figure]

The representativity of the resulting monthly long-term AOD record should be evaluated by comparison to other available multiyear records such as MODIS and MiSR (2000-present), SeaWIFS (1997-2010) and TOMS (1979-2001).

The annual means of different AOD data (the constructed AOD, OMI, MODIS, SeaWIFS and MISR) from the year 2010 are shown in figure below. The constructed AOD have higher values (around 0.25) compared to the MODIS, SeaWIFS and MISR AOD values in the North-America Taiga area, in the Savanna area in South-America, in the northern Europe, in the Siberia, in the central and eastern Asia area, in the east and southern Africa, and in the islands in the southeast Asia. On the other hand, the constructed AOD do not reach the level of the highest AOD values in the MODIS, SeaWIFS and MISR AOD maps in the Sahara, in the Savanna area in Africa and in the eastern China. In other areas the constructed AOD is close with the other AOD data. Overall the calculated AOD values have the similar behaviour as the MODIS, SeaWIFS and MISR AOD data, but are in some areas too high.

[Figure]

References

Ahn, C., O. Torres, and H. Jethva(2014), Assessment of OMI near-UV aerosol optical depth over land, J. Geophys. Res. Atmos., 119, 2457–2473, doi:10.1002/2013JD020188.

Carboni,E., et al.: Desert dust satellite retrieval intercomparison, Atmos. Meas. Tech. Di scuss., 5, 691-746, doi:10.5194/amtd-5-691-2012, 2012.

Kiss P., I.M. Janosi, and O. Torres, Early calibration problems detected in Earth Probe TOMS aerosol signal, Geophys. Res. Letters, 34 L07803, doi: 10.1029/2006GL028108, 2007.

Penning de Vries, M. and Wagner, T.: Modelled and measured effects of clouds on UV Aerosol Indices on a local, regional, and global scale, Atmos. Chem. Phys., 11, 12715-12735, doi:10.5194/acp-11-12715-2011, 2011.

Torres, O., A. Tanskanen, B. Veihelman, C. Ahn,R. Braak, P. K. Bhartia, P. Veefkind, and P. Levelt, Aerosols and Surface UV Products from OMI Observations: An Overview, , J. Geophys. Res.,112, D24S47, doi:10.1029/2007JD008809, 2007

---

## Author Comment (AC1)

Author's response to Interactive comments of Anonymous Referee #1 on "An Aerosol Optical Depth time series 1982–2014 for atmospheric correction based on OMI and TOMS Aerosol Index" by E. Jääskeläinen et al.

We thank the referee for careful reading of our manuscript and for the helpful comments. We will incorporate these comments to the revised manuscript. Below, we list referees' comments followed by our answers (in blue). The pages and lines included in our answers refer to the revised manuscript.

Atmospheric correction is needed to create land surface albedo data sets, and requires AOD. For the CLARA-A2-SAL albedo data set created from AVHRR measurements, for the time period 1982-2014, there is no available AOD product. However, UV aerosol index (AI) is available. This study presents a method to relate AI to AOD, and therefore provide an atmospheric correction for use in CLARA-A2-SAL processing. This is an improvement over the last version of the albedo data set, which used a constant AOD, which would therefore introduce some regional/seasonal biases into the albedo. The research is important and the basic premise is sound. However I have some concerns about some aspects of the analysis, and some parts are unclear. My scientific comments are as follows:

MODIS AOD is used to filter the OMI AI. However, from Table 2, it seems like 3 different versions of the MODIS AOD data were used for different parts of the time period. This is somewhat surprising to me since the current version, Collection 006, has been available for about 3 years now, and the differences between the data versions have been documented to be large in some versions. So this would potentially introduce some discontinuities in the data set. I can't think of a good reason for using multiple versions of the MODIS data since all the data are available freely and it should not be too much of a burden to obtain the latest versions. I therefore strongly recommend that the analysis is repeated using consistently the latest data version (C006), rather than a mixture of this and older obsolete versions.

We agree, the analysis should have been done using the MODIS data from the Collection 006, but the data set was not available when the constructing of the AOD time series began. The AOD time series presented in the manuscript have already been used for correcting the atmospheric input in the CLARA-A2-SAL product, and hence this version of the AOD time series cannot be changed. In the future, for the next versions of the AOD time series, this matter will be taken into account. Note of this is in the new section in the manuscript, section 7 titled as Discussion and conclusions.

It is also not quite clear to me what time period of MODIS data are used. Section 3 suggests that only the time period 2005-2008 was used, but Table 2 gives different data versions for different time windows. This should be clarified. Whichever period is used, the latest MODIS data should be used.

MODIS-AOD data are used for the time period 2005-2014. This detail is now clarified in the manuscript P. 10/L. 19. The process of generating the AOD time series was so time consuming that new versions of MODIS-AOD data appeared afterwards. This is inevitable in dealing with large data sets: whenever one has finalized one's own time series, some of the input providers have already improved their data sets. This is a magic circle and one just has to accept that.

On a related note, the authors don't say which MODIS AOD data product they are using (Dark Target, Deep Blue, or a combination). This should be stated. Both have advantages and limitations. For example Dark Target gives no coverage over deserts (Deep Blue does), while Dark Target has better coverage over tropical forests. Collection 006 contains a combined data set from both algorithms, which may be optimal here. Otherwise there will be lots of data coverage gaps. However in Figure 9, there is data over deserts, so perhaps Deep Blue is used. But it is not stated anywhere in the manuscript. And if not, then how is the AOD-AI regression done without MODIS AOD data over these regions, since section 3.4 says the regression is pixel-wise?

The AOD data from Dark Target and Deed Blue aerosol retrieval algorithms are used. This detail is now clarified in the manuscript P. 5/L. 12-13.

In section 3.1 it is not clear exactly how the OMI 550 nm AOD is created – specifically, the paper does not say where the Angstrom exponent is obtained from.  It sounds like the AOD is estimated from each wavelength (with an Angstrom exponent from an unknown source), and then the estimates from each of the 5 wavelengths propagated to 550 nm are averaged. Is that right? Would a better way not be to use all 5 wavelengths together with the Angstrom power law to derive AOD at 550 nm and Angstrom exponent in a self-consistent way?

The Ångström exponent is calculated once from each wavelength pair (ten combinations altogether) and these exponents are then used to estimate the AOD at 550 nm by using the AOD values at suitable wavelengths (again, ten AOD values at 550 nm altogether). The final AOD at 500 nm is then the mean value of these estimates. This is about what the reviewer suggests to do. Text updated P. 5/L. 6-9.

Section 3.3, I don't think that deseasonalisation of the AOD and AI data makes physical sense, and I am concerned that this will in fact introduce regional and seasonal artefacts into the data. AI depends on aerosol composition (amount and degree of absorption), altitude, and the underlying surface. This can vary widely from season to season within a given location. For example, patterns of biomass burning and dust aerosol tend to be highly seasonal. Vegetation phenology gives pronounced changes in the underlying surface cover, and seasonal differences in temperature and aerosol sources affect the aerosol height. All of these will modulate the AOD-AI relationship, and so there will be seasonally-dependent relationships in many regions. Yet as the authors note, the deseasonalisation step produces a seasonally-independent relationship. This will therefore introduce artefacts. For example, it could be responsible for some of the discrepancies in Figure 21.

This seems a misunderstanding. The seasonality is imported back (P. 8/L. 22-23 + P. 9/ L. 21-22) when the AOD time series are calculated from the OMI-AI and TOMS-AI data. It is now clarified in the manuscript, P. 6/L. 17.

Equation 5, what exactly are the 'modified' AOD and AI here? I did not find a definition for how these are different from the normal AOD and AI. Or does this refer to the deseasonalised data? This should be made clear.

The modified AOD and AI means preprocessed and deseasonalized AOD and AI data. This detail is now clarified in the manuscript P. 7/L. 11.

Table 2 also shows that the wavelengths that AI is calculated from differ between Nimbus-7 TOMS, Earth Probe TOMS, and OMI. Since UV aerosol extinction and absorption exhibit spectral dependence, these wavelength differences mean that AI calculated for the same aerosol would differ between the sensors. This effect was not really discussed but should ideally be quantified.

We apologize, there were errors in Table 2, the wavelength range for the calculation of the AI in Nimbus-7 is also 331-360 nm, the same as it is in Earth Probe, and in the OMI it is 354-388 nm. It is now corrected in the manuscript Table 2.

The UV AI is also only sensitive to aerosols which are light-absorbing in the UV. So weakly-absorbing or nonabsorbing aerosols, such as sulphates, will have no AI signal. Yet they will contribute to the AOD and so affect the atmospheric correction. The regression may account for this, to an extent – I suppose it will contribute to the term beta in Equation 5. However if the loading of nonabsorbing aerosols is variable in time, then this variation can't be captured by beta. The partition between nonabsorbing and absorbing aerosols in the AOD to AI conversion could be discussed in a bit more detail. Or is the pixel just completely discarded if the AI is too low, and not used to estimate albedo at all? This is hinted at in sections 2.2 and 3.1, but could be stated more explicitly.

The AI values below 0.5 are discarded completely from the analyses and from the AOD time series calculation, because we want to keep the constructed AOD time series homogeneous. This detail is mentioned in the following pages and lines:
Subsection 2.2: P. 4/L. 2-5,
Subsection 3.1: P. 5/L. 16-18,
Subsection 4.1: P. 8/L. 15-17

The presented method, whereby AOD is estimated from AI, is clearly a better assumption than taking a constant value of AOD as was done in the first version of the albedo data set. However, in light of the above issues, I recommend that the manuscript is revised and re-reviewed after the above aspects have been clarified. Then it will be easier to understand the subtleties of what is done.